# From Parameter Dynamics to Risk Scoring:
# Quantifying Sample-Level Safety Degradation in LLM Fine-tuning

Xiao Wang [1]   Yifei Zhang [† 1]   YongKang Liu [2]   Xiaocui Yang [1]   Zihan Wang [1]   Shi Feng [1]   Daling Wang [1]

## Abstract

Safety alignment of Large Language Models (LLMs) is extremely fragile, as fine-tuning on a small number of benign samples can erase safety behaviors learned from millions of preference examples. Existing studies attempt to explain this phenomenon by comparing parameters and hidden states before and after fine-tuning, but overlook their dynamic evolution during fine-tuning. In this paper, we uncover a critical mechanism underlying safety degradation by analyzing parameter dynamics, where benign fine-tuning causes parameters to cumulatively drift toward danger-aligned directions, progressively undermining the model's safety. This finding suggests that samples contributing more to this drift has greater fine-tuning risks. Based on this insight, we propose a method of Sample-Level Quantification of Safety Degradation (SQSD), which quantifies the influence of each training sample on safety degradation. Specifically, SQSD computes continuous risk scores to samples by measuring their induced parameter updates' projection difference between danger and safety directions. Extensive experiments across multiple models and datasets demonstrate that SQSD effectively quantifies sample-level fine-tuning risks and exhibits strong transferability across model architectures, parameter scales, and parameter-efficient methods[1].

## 1. Introduction

LLMs are widely deployed across real-world applications (Achiam et al., 2023; Lu et al., 2025; Jeong, 2024)

and are routinely adapted to downstream domains via post-training (Ouyang et al., 2022; Rafailov et al., 2023). Before deployment, model developers typically apply safety alignment to curb unsafe behaviors (Yang et al., 2025; Dubey et al., 2024). However, recent studies (Ji et al., 2025b; Lin et al., 2025; Qian et al., 2024; Bai et al., 2022; Ji et al., 2023) highlight the fragility of such alignment. More critically, even fine-tuning on merely 100 benign samples can severely degrade model safety (Qi et al., 2023; Guan et al., 2025; He et al., 2024; Zhan et al., 2024). Unlike explicitly harmful content that can be screened out by toxicity detection tools (Lees et al., 2022; Llama Team, 2024), benign samples can evade detection entirely, which make this form of safety degradation much harder to prevent than attacks using overtly harmful data. Motivated by the observation that benign fine-tuning can degrade model safety, we pose first research question:

> **RQ1: Why does fine-tuning on benign data lead to safety degradation?**

Previous studies have examined why fine-tuning degrades model safety by analyzing embedding drift (Huang et al., 2024) and parameter perturbations (Peng et al., 2024; Huang et al., 2025). However, these studies have two limitations: (1) analyzing only pre- and post-fine-tuning states fails to capture the dynamic evolution of these states during the fine-tuning process; (2) focusing on perturbation magnitude without considering directions changes affecting degradation capabilities, making it difficult to isolate safety-specific degradation. Meanwhile, these limitations prevent them from capturing the directional parameter drift that drives safety degradation during benign fine-tuning. Therefore, we study the parameter dynamics by tracking the trajectory of parameters throughout training and analyzing their alignment with safety-related directions. Through systematic analysis of parameter dynamics, a critical mechanism underlying safety degradation is revealed: benign fine-tuning will induce cumulative parameter drift toward danger directions, as shown in Figure 1(a). This provides a mechanistic explanation for RQ1.

Our finding raises a further discussion, where safety degradation stems from cumulative parameter drift, yet different samples contribute unequally to this drift. Some will ac-

---

[1]School of Computer Science and Engineering, Northeastern University, Shenyang, China [2]School of Computer and Communication Engineering, Northeastern University, Qinhuangdao, China. Correspondence to: Xiao Wang <wx191038@gmail.com>, Yifei Zhang <zhangyifei@cse.neu.edu.cn>.

*Proceedings of the 43rd International Conference on Machine Learning*, Seoul, South Korea. PMLR 306, 2026. Copyright 2026 by the author(s).

[1]Code is available in FinetuningRisk-Quantifying

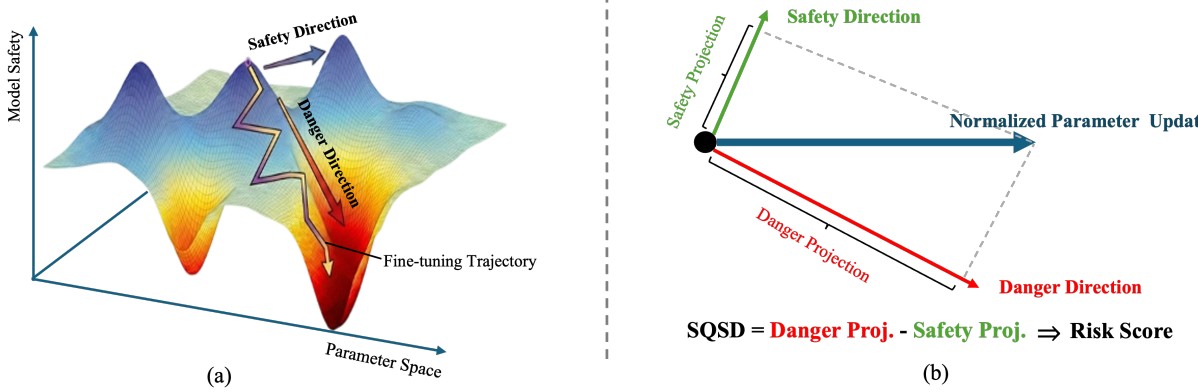

*Figure 1.* Overview of safety degradation mechanism and SQSD. **(a)**: Fine-tuning trajectory shows cumulative parameter drift toward danger-aligned direction in parameter space. **(b)**: SQSD computes risk scores by measuring the projection gap between sample-induced parameter updates and safety-relevant directions. Larger danger projection minus safety projection indicates higher risk.

celerate it substantially, while others have minimal impact. This motivates our second research question:

> **RQ2: To what extent does a training example steer the model toward the dangerous state?**

Existing research (He et al., 2024; Guan et al., 2025; Li et al., 2025b) attempts to answer this question by scoring benign samples and identifying high-risk subsets. This extreme sample selection approach suffers from the boundary collapse problem, where training only on extreme samples causes the model to learn discrete trigger patterns instead of a continuous risk perception function, resulting in poor generalization to samples with intermediate risk levels. To address this limitation, we propose a method of **SQSD** (**S**ample-Level **Q**uantification of **S**afety **D**egradation), which computes continuous risk scores to every sample in the corpus, enabling fine-grained risk assessment across the entire risk spectrum rather than discrete subset selection. Using the parameter dynamics view, SQSD quantifies the extent to which a training example drives the model toward dangerous states and away from safe ones by measuring the projection difference of its induced parameter updates along danger versus safety directions, as shown in Figure 1(b), which directly responds RQ2. It is crucial that SQSD executes on parameter space, enabling it to quantify fine-tuning risks for benign samples that evade traditional toxicity classifiers. In addition, we provide a theoretical foundation by first-order Taylor approximation, which links SQSD to preference differences between safe and unsafe models and offers an interpretable connection between parameter update and model behavior. Our contributions are summarized as follows:

- We reveal the dynamical mechanism underlying safety degradation by tracking parameter trajectories during benign fine-tuning: parameters cumulatively drift toward danger-aligned directions, progressively under-

mining safety.

- We propose the SQSD method, which quantifies each benign training example's fine-tuning risk for safety degradation through directional analysis of parameter updates. In particular, SQSD provides continuous risk quantification across the entire training corpus.

- Through extensive experiments across three models and two datasets, SQSD demonstrates its excellent performance on quantifying sample-level risks of fine-tuning. It also shows strong transferability across model architectures, parameter scales, and parameter-efficient methods (from LoRA to Full Fine-tuning).

## 2. Related Works

### 2.1. Safety Degradation in Fine-tuning

Before releasing LLMs (Yang et al., 2025; Dubey et al., 2024), model providers typically apply safety alignment methods of post-training, such as RLHF (Ouyang et al., 2022; Bai et al., 2022) and DPO (Rafailov et al., 2023), to ensure models refuse harmful requests. However, recent studies reveal that safety alignment is fragile and can be compromised during fine-tuning. This occurs not only on fine-tuning with explicitly harmful examples (Lin et al., 2025; Qian et al., 2024; Ji et al., 2025b), but more alarmingly, even with benign samples from standard instruction-tuning datasets (Qi et al., 2023; Eiras et al., 2025; Zhan et al., 2024; He et al., 2024; Guan et al., 2025). Unlike explicitly harmful examples that toxicity detectors (Lees et al., 2022; Llama Team, 2024) can filter out, benign samples leading to safety degradation are usually harder to prevent since they can evade filtering detection.

### 2.2. Safety Degradation Mechanism

Recent studies examine why benign fine-tuning degrades safety from multiple perspectives. From **model-centric**

**perspective**, alignment operates superficially confined to early output tokens (Qi et al., 2025) and sparse parameter regions (Wei et al., 2025), and the "elasticity" effect (Ji et al., 2025b) causes rapid regression during fine-tuning due to pre-training data dominance. From **data-centric perspective**, similarity between fine-tuning and alignment (Hsiung et al., 2025) and specific content features (Li et al., 2025a; Pandey et al., 2025) can override safety alignment. From **parameter-level perspective**, aligned models occupy a "safe basin" (Peng et al., 2024) in parameter space, but downstream optimization will displace parameters when task and safety optima diverge (Chen et al., 2025), and this situation will be amplified by harmful perturbation patterns(Huang et al., 2025) and embedding drift (Huang et al., 2024). Previous works examine these phenomena through static parameter perturbations (Huang et al., 2025; Wei et al., 2025) or embedding analysis (Huang et al., 2024), while we track cumulative parameter drifts to danger directions on training, and provide a dynamic perspective on safety degradation.

## 2.3. Sample-Level Influence and Risk Quantification

Exploring individual training samples' influence on model behavior is a longstanding challenge. Previous studies estimate sample influence through gradient-based methods (Pruthi et al., 2020; Xia et al., 2024) and learning dynamics analysis (Ren & Sutherland, 2025). Recent work extends to safety implications during benign fine-tuning. Bi-Anchor (He et al., 2024) use bi-directional anchoring in representation and gradient space to identify high-risk samples. Self-Inf-N (Guan et al., 2025) demonstrates that gradient-based outlier benign samples disproportionately break alignment. LARF (Li et al., 2025b) identifies high-risk samples through representation similarity at safety-sensitive layers. Detailed descriptions are provided in Appendix D. However, they identify discrete harmful subsets rather than providing continuous, corpus-wide risk assessments.

## 3. Safety Degradation in Parameter Dynamics

### 3.1. Preliminaries

**Parameter Drift with LoRA.** We characterize parameter changes induced by LoRA fine-tuning (Hu et al., 2022). Consider a linear module with base weight $W \in \mathbb{R}^{d_{\text{out}} \times d_{\text{in}}}$, LoRA augments it with low-rank matrices $A \in \mathbb{R}^{r \times d_{\text{in}}}$ and $B \in \mathbb{R}^{d_{\text{out}} \times r}$, yields the effective weight:

$$W' = W + \frac{\alpha}{r} BA, \qquad (1)$$

where $r$ is the rank and $\alpha$ is a scaling factor. The *parameter drift* for this module is defined as:

$$\Delta W \triangleq W' - W = \frac{\alpha}{r} BA. \qquad (2)$$

For $M$ LoRA-augmented modules of a model, e.g., attention projections $\{q, k, v, o\}$ and feed-forward projections $\{gate, up, down\}$, let $\Delta \theta = \{\Delta W_1, \ldots, \Delta W_M\}$ denote the collection of all parameter drifts. These parameter drifts enable us to construct safety-relevant directions and track parameter trajectories throughout fine-tuning.

**Safety and Danger Directions.** We define two reference directions in parameter space: the *safety direction* $V_{\text{safety}}$ and the *danger direction* $V_{\text{danger}}$. These directions serve as semantic anchors for analyzing safety degradation and quantifying sample-level risk.

Following the Task Vector formulation (Ilharco et al., 2023), these directions are constructed as parameter displacements from a base model $\theta_0$ to safety-aligned and harm-aligned states:

$$V_{\text{safety}} = \hat{\theta}_{\text{aligned}} - \theta_0,$$
$$\hat{\theta}_{\text{aligned}} = \arg\min_\theta L_{\text{dpo}}(\theta_0, D_{\text{aligned}}), \qquad (3)$$

$$V_{\text{danger}} = \hat{\theta}_{\text{harmful}} - \theta_0,$$
$$\hat{\theta}_{\text{harmful}} = \arg\min_\theta L_{\text{sft}}(\theta_0, D_{\text{harmful}}), \qquad (4)$$

where $\hat{\theta}_{\text{aligned}}$ is obtained by applying Direct Preference Optimization on PKU-SafeRLHF-10K (Ji et al., 2023) for the safety direction, while $\hat{\theta}_{\text{harmful}}$ is obtained via Supervised Fine-Tuning on Aegis (Ghosh et al., 2024) and Beaver-Tails (Ji et al., 2023) for the *Aegis-unsafe* and *Beaver-unsafe* danger directions, respectively. Complete construction details are provided in Appendix A.1.

**Direction Verification.** To verify that these directions capture safety-relevant behavioral changes, we perform parameter steering experiments on the initial model $\theta_0$. Specifically, we linearly perturb the model parameters along each direction and measure the resulting safety changes:

$$\theta(\alpha) = \theta_0 + \alpha V, \qquad V \in \{V_{\text{safety}}, V_{\text{danger}}\}, \qquad (5)$$

where $\alpha$ is the steering magnitude that controls the strength of the directional perturbation. We evaluate the safety of the steered model $\theta(\alpha)$ across different values of $\alpha$ to examine whether model safety changes consistently with the variation of $\alpha$. As detailed in Appendix A.2, steering along $V_{\text{danger}}$ consistently decreases model safety as $\alpha$ increases, while steering along $V_{\text{safety}}$ exhibits the opposite trend. These monotonic relationships between $\alpha$ and safety performance confirm that our defined directions reliably encode safety-relevant parameter displacements.

### 3.2. Safety Degradation Analysis

From the parameter dynamics perspective, we investigate the mechanism underlying safety degradation during benign fine-tuning. Rather than viewing fine-tuning as a single parameter perturbation (Huang et al., 2025; Peng et al.,

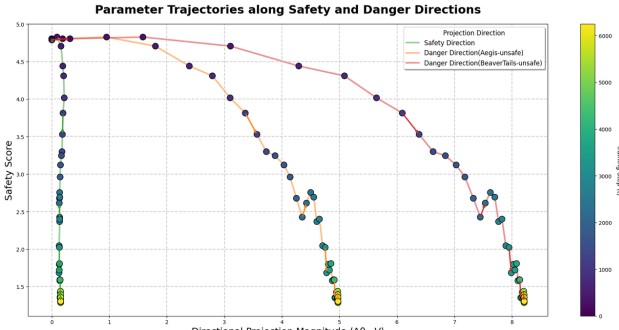

*Figure 2.* Parameter Drift trajectories along safety and danger directions during fine-tuning. Qwen3-8b fine-tuned Dolly (5k). Safe Score is a safety metric (higher is safer); $\langle \Delta\theta, V \rangle$ is projection of parameter drift onto each direction. Safety-related directions details are provided in §3.1

2024), we track parameter trajectories during fine-tuning and link their directional drift to changes in safety behavior.

**Tracking Parameter Drift via Directional Projection.** To characterize how model parameters evolve along safety-critical directions during fine-tuning, we track parameter drift at each training step and project it onto the safety and danger directions. Let $\theta_t$ denote the model parameters at training step $t$, the cumulative parameter drift from the initial model $\theta_0$ is:

$$\Delta\theta_t = \theta_t - \theta_0. \qquad (6)$$

The alignment between this drift and safety-relevant directions is measured via directional projections:

$$p_{\text{safety}}(t) = \langle \Delta\theta_t, \hat{V}_{\text{safety}} \rangle, \qquad (7)$$

$$p_{\text{danger}}(t) = \langle \Delta\theta_t, \hat{V}_{\text{danger}} \rangle, \qquad (8)$$

where $\hat{V} = V/\|V\|_2$ denotes the normalized direction.

**Parameter Dynamics During Fine-tuning.** Figure 2 shows the parameter drift trajectories and corresponding Safety Scores during fine-tuning of **Qwen3-8B** (Yang et al., 2025) on **Dolly**(5k) (Conover et al., 2023). Safety Score is a reward-model-based metric that quantifies the overall safety of model responses (detailed in Appendix B). This configuration demonstrates the core pattern of safety degradation. Its generality across diverse models, datasets, and scales is validated in § 5.2. As shown in Figure 2, the results show a consistent pattern: the projections of parameter drift onto both danger directions, including *Aegis-unsafe* and *Beaver-unsafe*, increase steadily throughout training, while the projection onto the safety direction remains near zero. Concurrently, this directional drift is accompanied by severe safety degradation, with the Safety Score declining from approximately 5.0 to below 1.0. This pattern provides a parameter dynamics explanation for RQ1: **fine-tuning on benign datasets induces cumulative drift of model parameters in danger-aligned directions**, progressively

eroding model safety. Notably, the trajectory exhibits a striking nonlinear phenomenon with two distinct phases: in the early stage, parameters drift rapidly in danger-aligned directions (projection magnitude increasing from 0 to 6.0) while safety degradation remains moderate (Safety Score declining from 5.0 to 4.0). Subsequently, despite the deceleration of directional drift, safety collapse accelerates dramatically, with the Safety Score plummeting from 4.0 to below 1.0. This asymmetric relationship indicates that (1) **safety degradation occurs predominantly after a substantial magnitude of parameter drift has accumulated in danger directions**; (2) **robustness to directional perturbations is local and limited**, consistent with the notion of a safety basin (Peng et al., 2024), the model tolerates moderate drift but degrades catastrophically once parameters exit this safe region.

To further substantiate the causal relationship between cumulative parameter drift along danger directions and safety degradation, rather than mere correlation, we provide additional analysis in Appendix I.

## 4. Sample-Level Risk Quantification

Parameter drift toward danger-aligned directions drives safety degradation. This observation suggests a natural hypothesis: *if a sample induces larger parameter updates along danger directions, training on it will cause more severe safety degradation*. Motivated by this intuition, we propose a method of SQSD, which quantifies each sample's fine-tuning risk by the projection gap of its induced parameter update along danger versus safety directions. Additionally, a theoretical connection between parameter displacement and output preferences is established via first-order Taylor approximation, and the role of model initialization in reliable SQSD computation is discussed.

### 4.1. SQSD

SQSD computes a sample's risk score in three steps: (1) compute the sample-induced parameter update via one-step gradient; (2) project this update onto danger and safety directions for each module; and (3) aggregate the projection gap across all modules.

**Sample-Induced Parameter Update.** We characterize the sample-induced parameter update through the gradients of LoRA parameters, where the LoRA weights are denoted by $A \in \mathbb{R}^{r \times d_{\text{in}}}$ and $B \in \mathbb{R}^{d_{\text{out}} \times r}$ with initial values $A_0$ and $B_0$. For a single training sample $z = (x, y)$, a one-step gradient descent (GD) update of the LoRA parameters takes the form:

$$\begin{aligned} \Delta A &= -\eta \, \nabla_{A_0} \mathcal{L}_{\text{sft}}(z), \\ \Delta B &= -\eta \, \nabla_{B_0} \mathcal{L}_{\text{sft}}(z), \end{aligned} \qquad (9)$$

where $\eta$ is the learning rate and $\nabla_{A_0|B_0} \mathcal{L}_{\text{sft}}(z)$ denotes the

gradients with respect to the LoRA parameters. The corresponding update to the LoRA-augmented weight is:

$$\Delta W(z) \approx B_0 \Delta A + \Delta B A_0 = -\eta \left( B_0 \nabla_A + \nabla_B A_0 \right),$$
(10)

where the second-order term $\Delta B \, \Delta A = \mathcal{O}(\eta^2)$ is negligible and thus omitted. This update captures the instantaneous parameter drift induced by sample $z$, we quantify the sample's risk by analyzing its alignment with safety-relevant directions.

**Module-wise Directional Projection and Aggregate.**
**Module-wise normalization** is first applied to the parameter updates before computing projections. For the $m$-th LoRA-augmented weight matrix, we compute the projection gap between the normalized update and the danger versus safety directions:

$$\mathrm{SQSD}_m(z) = \left\langle \frac{\Delta W_m(z)}{\|\Delta W_m(z)\|_2}, \hat{V}_{\mathrm{danger},m} \right\rangle$$
$$- \left\langle \frac{\Delta W_m(z)}{\|\Delta W_m(z)\|_2}, \hat{V}_{\mathrm{safety},m} \right\rangle,$$
(11)

Here, $\hat{V} = V/\|V\|_2$ denotes $L_2$-normalized direction vectors and $\langle \cdot, \cdot \rangle$ denotes the inner product. Finally, the projection gaps are aggregated across all LoRA-augmented modules to obtain the final SQSD score:

$$\mathrm{SQSD}(z) = \sum_m \mathrm{SQSD}_m(z).$$
(12)

Previous gradient-based scoring methods (He et al., 2024; Guan et al., 2025) are known to exhibit response-length bias when using raw or unnormalized updates. The same effect is observed here: shorter-response examples tend to obtain higher scores when $\Delta W_m(z)$ is not normalized, despite not always contributing more to safety degradation. We thus adopt module-wise normalization for $\mathrm{SQSD}(z)$ and defer further analysis to Appendix F. In parameter space, $\mathrm{SQSD}(z)$ quantifies the directional preference of the sample-induced update by comparing its alignment with $V_{\mathrm{danger}}$ and $V_{\mathrm{safety}}$. A larger $\mathrm{SQSD}(z)$ indicates that updating on $z$ moves the parameters more toward the dangerous parameter state than toward the safe one, whereas a smaller (or negative) score indicates the opposite.

### 4.2. Connecting SQSD to Output Preferences

Following prior work (He et al., 2024; Ren & Sutherland, 2025), we use the **first-order Taylor approximation** to relate the inner product between a sample-induced update and a displacement direction to the corresponding loss change. This provides a preference-based interpretation of SQSD.

Consider a training sample $z = (x, y)$ and two parameter states $\theta_{\mathrm{ref}}$ and $\theta_{\mathrm{target}}$ (an initial model and its fine-tuned counterpart). Under first-order taylor approximation (derivation

in Appendix C),

$$\eta \Big[ \mathcal{L}(z, \theta_{\mathrm{ref}}) - \mathcal{L}(z, \theta_{\mathrm{target}}) \Big] \approx (\theta' - \theta_{\mathrm{ref}})^\top (\theta_{\mathrm{target}} - \theta_{\mathrm{ref}}),$$
(13)

where $\theta'$ denotes the parameters after a single gradient step on $z$ from $\theta_{\mathrm{ref}}$ with learning rate $\eta$.

**Interpretation of Loss Difference.** Eq. (13) links an inner product in parameter drift to the corresponding change in loss. Under token-level Negative Log-Likelihood, lower loss corresponds to higher conditional likelihood $p_\theta(y \mid x)$, meaning that $(x, y)$ is more consistent with the model's output preference under $\theta$. Define $\Delta \mathcal{L}_{\mathrm{ref} \to \mathrm{target}}(z) = \mathcal{L}(z, \theta_{\mathrm{ref}}) - \mathcal{L}(z, \theta_{\mathrm{target}})$. Given $x$, a larger positive $\Delta \mathcal{L}_{\mathrm{ref} \to \mathrm{target}}(z)$ indicates that $\theta_{\mathrm{target}}$ assigns higher likelihood to $y$ than $\theta_{\mathrm{ref}}$. Equivalently, updating on sample $z$ from $\theta_{\mathrm{ref}}$ pushes the model parameters toward $\theta_{\mathrm{target}}$, as evidenced by the positive inner product between the induced update and the displacement $(\theta_{\mathrm{target}} - \theta_{\mathrm{ref}})$.

**Connection to SQSD.** In our setting, instantiating $\theta_{\mathrm{target}}$ as $\theta_{\mathrm{danger}}$ or $\theta_{\mathrm{safety}}$ yields two loss differences: $\Delta \mathcal{L}_{\mathrm{ref} \to \mathrm{danger}}(z)$ and $\Delta \mathcal{L}_{\mathrm{ref} \to \mathrm{safety}}(z)$. By Eq. (13), loss differences are approximated by the corresponding inner products between the sample-induced update and the two directions. Since SQSD computes the gap between these inner products (equivalently, the difference between the two loss changes), a larger SQSD indicates that updating on $z$ from $\theta_{\mathrm{ref}}$ steers parameters more toward $\theta_{\mathrm{danger}}$ than toward $\theta_{\mathrm{safety}}$ in parameter space, meaning sample $z$ is better aligned with the danger state than the safety state. Conversely, a smaller or negative score indicates the update favors the safety direction. Thus, SQSD directly links parameter updates to safety behaviors.

### 4.3. Parameter Initialization

SQSD computes the projection gap between a sample-induced parameter update and two safety-relevant directions, where the update is derived from instantaneous gradients while the directions capture cumulative parameter drift from complete fine-tuning runs. Different parameter states exhibit different directional sensitivities, and the same perturbation can induce vastly different safety changes at different parameter states, as evidenced by the nonlinear dynamics in Figure 2 and the non-uniform sensitivity across $\alpha$ in Figure 5, demonstrating that SQSD's effectiveness is parameter-dependent. Therefore, we initialize at parameter states exhibiting high directional sensitivity to ensure reliable risk quantification.

We formalize directional sensitivity under two scenarios. **Linear-path sensitivity** ($\theta_{\mathrm{initial}} = \theta_0 + \alpha V$) measures how safety changes along the interpolation path, while **drift-enhanced sensitivity** ($\theta_{\mathrm{initial}} = \theta_t$ during fine-tuning) captures sensitivity after cumulative directional drift. Based on their distinct geometric properties, we initialize $\theta_{\mathrm{safety}} =$

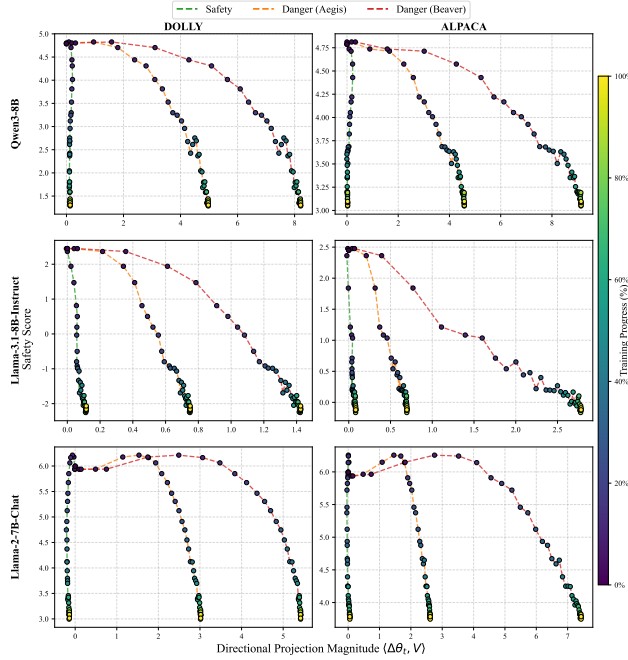

Figure 3. Consistency of parameter-space mechanism across models and datasets. Parameter trajectories along safety and danger directions for three models (Llama-3.1-8B-Instruct, Qwen3-8B, Llama-2-7B-Chat) fine-tuned on 5k-Dolly and 5k-Alpaca.

$\theta_0 + \alpha^* V_{\text{safety}}$ (selecting $\alpha^*$ from locally sensitive ranges) and $\theta_{\text{danger}} = \theta_{t^*}$ (selecting high-sensitivity checkpoints from fine-tuning). Complete formulations and procedures are in Appendix E.

## 5. Experiments

### 5.1. Experimental Setups

**Models.** Three safety-aligned models are used for main experiments: Qwen3-8B (Yang et al., 2025), LLaMA-3.1-8B-Instruct (Dubey et al., 2024), and LLaMA-2-7B-Chat (Touvron et al., 2023). For cross-scale transferability (§5.3.2), we employ Qwen3-14B and Qwen3-32B.

**Datasets.** Two categories of datasets are used: benign fine-tuning data and direction construction data. For benign fine-tuning, Alpaca (Taori et al., 2023) and Dolly (Conover et al., 2023) are used, with 5k samples by default unless otherwise specified. For direction construction, PKU-SafeRLHF-10k (Ji et al., 2023) is used for $V_{\text{safety}}$, and 3k samples from the unsafe subsets of Aegis (Ghosh et al., 2024) and Beaver-Tails (Ji et al., 2023) for $V_{\text{danger}}$.

**Safety Evaluation.** Evaluation is conducted three safety benchmarks: CategoricalHarmfulQA (Bhardwaj et al., 2024), AdvBench (Zou et al., 2023) and HEx-PHI (Qi et al., 2023), reporting CategoricalHarmfulQA results by default. We use ASR (with LlamaGuard3-8B (Llama Team, 2024)) and Safety Score (with `beaver-7b-unified-cost` (Ji

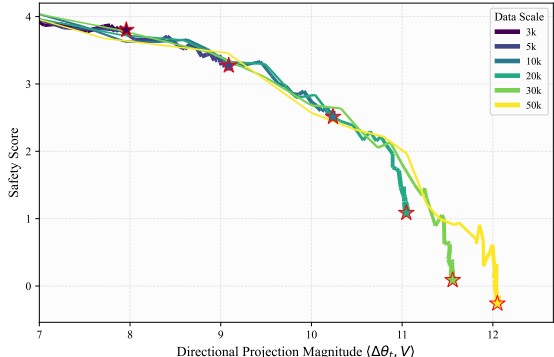

Figure 4. Impact of dataset scale on parameter drift. Trajectories for Qwen3-8B on 3k–50k Alpaca samples.

et al., 2025a)) as metrics, more details in Appendix B. All responses use greedy decoding.

**Training Configuration.** All benign fine-tuning uses LoRA (Hu et al., 2022) ($r = 8$, $\alpha = 16$) with AdamW, batch size 8, 10 epochs, via LLaMA-Factory (Zheng et al., 2024). Learning rate is $5 \times 10^{-6}$ for mechanism validation to produce smoother parameter trajectories, and $5 \times 10^{-5}$ for SQSD evaluation to induce stronger safety degradation. For full fine-tuning in transferability experiments, $5 \times 10^{-6}$ is used as it requires smaller learning rates than LoRA. Direction construction and validation is detailed in Appendix A.

**Baselines.** We compare SQSD with existing sample-level influence methods: Bi-Anchor(Reps/Grad) (He et al., 2024), Self-Inf-N (Guan et al., 2025), and LARF (Li et al., 2025b). We also include Reward Model (Ji et al., 2025a) as a natural baseline that directly scores sample safety using a pretrained reward model. Implementation details are in Appendix D.

### 5.2. Validation of the Cumulative Drift Mechanism

The parameter dynamics mechanism (§3.2) is validated across three models (Qwen3-8B, Llama-3.1-8B-Instruct, Llama-2-7B-Chat), two datasets (Dolly, Alpaca), and multiple data scales (3k–50k samples).

**Consistency Across Models and Datasets.** As shown in Figure 3, the consistent pattern across all six configurations is observed: projections onto both *Aegis-unsafe* and *Beaver-unsafe* danger directions increase monotonically throughout training, while safety projections remain near zero or slightly negative. This directional drift consistently correlates with declining Safety Scores. However, the dynamics of safety degradation exhibit notable model-specific characteristics. Qwen3-8B consistently displays a pronounced two-phase degradation pattern across both datasets, where Safety Score declines gradually in early training before collapsing rapidly in later stages. Llama-2-7B-Chat shows similar two-phase behavior on both datasets, while Llama-3.1-8B-Instruct exhibits two-phase degradation on Dolly but

*Table 1.* Effectiveness of SQSD. ASR (%) on CategoricalHarmfulQA for models fine-tuned on risk-ranked subsets by various methods. S1-S5 represent 1000 samples each, uniformly sampled from highest to lowest risk rankings. Δ denotes ASR difference (S1 - S5). Mono indicates whether ASR decreases monotonically across subsets (✓ represents yes; ✗ is No.)

| Method | Dolly | | | | | | | Alpaca | | | | | | |
|---|---|---|---|---|---|---|---|---|---|---|---|---|---|---|
| | S1 | S2 | S3 | S4 | S5 | Δ↑ | Mono | S1 | S2 | S3 | S4 | S5 | Δ↑ | Mono |
| *Qwen3-8B* | | | | | | | | | | | | | | |
| Reward Model | 59.82 | 14.91 | 4.55 | 13.82 | 5.45 | 54.37 | ✗ | 50.73 | 6.00 | 9.09 | 9.82 | 6.91 | 43.82 | ✗ |
| Bi-Anchor(Reps) | 9.64 | 16.36 | 52.18 | 27.64 | 25.64 | -16.00 | ✗ | 8.73 | 15.27 | 14.73 | 5.45 | 2.55 | 6.18 | ✗ |
| Bi-Anchor(Grad) | 3.45 | 28.91 | 38.73 | 22.36 | 15.64 | -12.19 | ✗ | 2.36 | 4.55 | 26.36 | 8.00 | 5.82 | -3.46 | ✗ |
| Self-Inf-N | 48.36 | 34.73 | 7.27 | 36.36 | 20.91 | 27.45 | ✗ | 2.91 | 1.45 | 7.09 | 7.09 | 25.82 | -22.91 | ✗ |
| LARF | 84.18 | 70.91 | 12.73 | 18.91 | 1.09 | **83.09** | ✗ | 13.82 | 22.36 | 4.36 | 1.09 | 2.00 | 11.82 | ✗ |
| **SQSD(Aegis)** | 45.27 | 28.36 | 15.45 | 4.55 | 2.73 | 42.54 | ✓ | 40.91 | 14.36 | 7.27 | 4.00 | 2.55 | 38.36 | ✓ |
| **SQSD(Beaver)** | 71.27 | 29.45 | 10.18 | 7.27 | 2.55 | 68.72 | ✓ | 50.91 | 19.09 | 18.73 | 7.27 | 3.27 | **47.64** | ✓ |
| *Llama3.1-8B-Instruct* | | | | | | | | | | | | | | |
| Reward Model | 67.64 | 43.27 | 18.55 | 37.82 | 20.18 | 47.46 | ✗ | 70.18 | 7.45 | 34.00 | 15.45 | 15.82 | 54.36 | ✗ |
| Bi-Anchor(Reps) | 46.36 | 37.09 | 40.55 | 42.00 | 23.09 | 23.27 | ✗ | 20.00 | 17.45 | 32.36 | 20.18 | 42.00 | -22.00 | ✗ |
| Bi-Anchor(Grad) | 56.55 | 50.36 | 35.64 | 22.18 | 16.91 | 39.64 | ✓ | 33.09 | 27.27 | 18.18 | 37.64 | 26.73 | 6.36 | ✗ |
| Self-Inf-N | 22.91 | 38.73 | 27.64 | 27.64 | 27.64 | -4.73 | ✗ | 77.27 | 32.55 | 31.82 | 56.36 | 38.91 | 38.36 | ✗ |
| LARF | 61.27 | 68.73 | 33.09 | 43.82 | 11.09 | 50.18 | ✗ | 64.73 | 35.27 | 44.36 | 12.55 | 4.91 | 59.82 | ✗ |
| **SQSD(Aegis)** | 76.36 | 21.45 | 36.36 | 16.18 | 13.82 | 62.54 | ✗ | 49.64 | 29.82 | 23.09 | 16.91 | 8.55 | 41.09 | ✓ |
| **SQSD(Beaver)** | 79.82 | 57.27 | 37.45 | 5.27 | 4.73 | **75.09** | ✓ | 65.82 | 27.27 | 25.45 | 9.64 | 0.36 | **65.46** | ✓ |
| *Llama2-7b-Chat* | | | | | | | | | | | | | | |
| Reward Model | 44.36 | 6.00 | 6.00 | 1.64 | 0.00 | 44.36 | ✓ | 18.55 | 3.82 | 1.45 | 0.00 | 0.36 | 18.19 | ✗ |
| Bi-Anchor(Reps) | 13.27 | 10.36 | 5.82 | 0.55 | 10.55 | 2.72 | ✗ | 10.91 | 14.00 | 3.27 | 0.73 | 1.45 | 9.46 | ✗ |
| Bi-Anchor(Grad) | 22.36 | 4.36 | 4.73 | 2.00 | 11.82 | 10.54 | ✗ | 5.27 | 3.64 | 1.45 | 1.64 | 6.00 | -0.73 | ✗ |
| Self-Inf-N | 3.45 | 4.55 | 0.91 | 7.09 | 1.64 | 1.81 | ✗ | 3.64 | 0.55 | 2.73 | 5.82 | 4.73 | -1.09 | ✗ |
| LARF | 1.27 | 3.45 | 15.82 | 6.55 | 1.64 | -0.37 | ✗ | 0.73 | 2.18 | 2.36 | 6.73 | 2.36 | -1.63 | ✗ |
| **SQSD(Aegis)** | 43.27 | 17.27 | 8.36 | 3.27 | 0.00 | 43.27 | ✓ | 39.27 | 3.27 | 1.09 | 0.55 | 0.36 | **38.91** | ✓ |
| **SQSD(Beaver)** | 45.27 | 14.73 | 5.45 | 2.55 | 0.36 | **44.91** | ✓ | 30.00 | 1.45 | 2.00 | 0.00 | 0.18 | 29.82 | ✗ |

more uniform degradation on Alpaca.

**Effect of Dataset Scale.** Figure 4 demonstrates that the parameter dynamics mechanism remains consistent across different data scales (3k–50k Alpaca samples). As expected, larger datasets induce both stronger cumulative drift toward danger directions and more severe safety degradation. The final directional projection magnitude increases monotonically with data scale: from approximately 8 (3k samples) to 12 (50k samples) along the Beaver-unsafe direction, with corresponding Safety Score deterioration from 3.8 to -0.3.

### 5.3. Main Result: Evaluation of SQSD

#### 5.3.1. EFFECTIVENESS VALIDATION

**Results.** SQSD's effectiveness is validated by partitioning each dataset into 5 subsets (from S1 to S5), where each subset has 1000 samples and is uniformly sampled across risk score rankings from highest to lowest, then fine-tuning separate models on each subset to measure the resulting

safety degradation. An effective risk quantification method should demonstrate two critical capabilities: (1) **consistent predictive power** for safety degradation severity, evidenced by monotonically decreasing ASR from S1 to S5 (Mono: ✓), and (2) **strong discriminative ability** between extreme risk, where high-risk subsets maximally degrade model safety while low-risk subsets cause negligible impact, resulting in a large Δ (ASR difference: S1 - S5). Table 1 presents the ASR on CatHarmfulQA for models fine-tuned on datasets sampled by different risk quantification methods, where SQSD(Beaver) and SQSD(Aegis) represent SQSD using two different danger directions. Results on other safety metrics and benchmarks are in Appendix J.4. As shown in Table 1, SQSD demonstrates consistent superiority across nearly all configurations. Models fine-tuned on SQSD-ranked subsets exhibit monotonically decreasing ASR in 10/12 settings, validating that SQSD effectively quantifies sample-level contributions to safety degradation across the entire corpus. In contrast, baselines fail to maintain monotonicity: Reward Model and Bi-Anchor(Grad)

achieves monotonicity in only 1/6 cases, and all other baselines in 0/6. Beyond monotonicity, SQSD demonstrates superior discriminative power in identifying samples with extreme risk for safety as measured by $\Delta$ (ASR difference: S1 - S5). Across all 12 configurations, SQSD consistently achieves the largest or near-largest $\Delta$, with an average of 49.86%, significantly exceeding the best baseline (Reward Model: 43.76%). This superior discrimination enables more precise data curation for safety fine-tuning.

To further demonstrate that SQSD quantifies sample-level fine-tuning risk across diverse safety categories rather than a single aggregate metric, we additionally report **category-wise ASR** on a benchmark spanning multiple safety categories in Appendix J.1. Additionally, the **comparison of computational efficiency** across methods and the generalization of SQSD to **domain-specific datasets** are provided in Appendix J.2 and J.3, respectively.

### 5.3.2. TRANSFERABILITY ANALYSIS

SQSD's transferability is evaluated across model architectures, parameter scales and parameter-efficient methods by computing risk scores under source configurations, partitioning datasets into five subsets, and fine-tuning models under target configurations on these subsets. Table 2 presents the resulting ASR, detailed configuration settings can be found in Appendix G. Despite substantial architectural differences between Llama3.1-8B-Instruct and Qwen3-8B, ASR decreases monotonically in both transfer directions (42.55%→1.64% and 79.64%→28.00%), demonstrating that SQSD captures architecture-agnostic sample-level risk. SQSD scores from Qwen3-8B also transfer robustly to larger variants (8B→14B: 55.09%→7.09%; 8B→32B: 28.91%→2.00%), enabling practitioners to compute risk scores on smaller models for larger deployment models. Furthermore, SQSD computed from LoRA gradients maintains discriminative power when transferred to full parameter fine-tuning (10.73%→2.55%). Across all three dimensions, SQSD consistently maintains monotonic rankings, confirming it captures fundamental sample-level characteristics underlying safety degradation.

### 5.3.3. ABLATION STUDIES

Ablation experiments are conducted using Qwen3-8B fine-tuned on Dolly with Beaver-unsafe direction. Results are shown in Table 3. **Module-wise normalization.** As described in Section 5.3, each module's update $\Delta W_m(z)$ is normalized to avoid response-length bias. The *w/o norm* variant computes SQSD using unnormalized updates, resulting in severe performance degradation ($\Delta$ drops from 68.72 to 12.54, monotonicity lost). This confirms that module-wise normalization effectively mitigates performance degradation caused by response-length bias. **Projection-gap**

*Table 2.* SQSD transferability across architectures (Qwen3-8B ↔ Llama3.1-8B-Instruct), parameter scales (8B→14B/32B), and parameter-efficient methods (LoRA→Full).

| Source → Target | S1 | S2 | S3 | S4 | S5 | Mono |
|---|---|---|---|---|---|---|
| *Architecture* | | | | | | |
| Llama→Qwen | 42.55 | 27.27 | 6.55 | 3.82 | 1.64 | ✓ |
| Qwen→Llama | 79.64 | 71.09 | 33.64 | 28.36 | 28.00 | ✓ |
| *Parameter Scale* | | | | | | |
| Qwen-8B→14B | 55.09 | 22.18 | 9.82 | 9.82 | 7.09 | ✓ |
| Qwen-8B→32B | 28.91 | 9.82 | 8.00 | 2.55 | 2.00 | ✓ |
| *PE Method* | | | | | | |
| Qwen(LoRA→Full) | 10.73 | 7.82 | 5.27 | 3.82 | 2.55 | ✓ |

*Table 3.* Ablation study on SQSD design choices.

| Method | S1 | S2 | S3 | S4 | S5 | $\Delta\uparrow$ | Mono |
|---|---|---|---|---|---|---|---|
| **SQSD (full)** | 71.27 | 29.45 | 10.18 | 7.27 | 2.55 | **68.72** | ✓ |
| *w/o norm* | 13.09 | 31.64 | 41.27 | 16.91 | 0.55 | 12.54 | ✗ |
| *Danger only* | 68.36 | 21.27 | 5.82 | 16.36 | 3.82 | 64.54 | ✗ |
| *Safety only* | 27.09 | 12.00 | 12.36 | 9.45 | 6.18 | 20.91 | ✗ |
| *Insens. init* | 38.36 | 28.00 | 10.00 | 10.91 | 1.09 | 37.27 | ✗ |

**design.** Using only danger direction (*Danger only*) or safety direction (*Safety only*) both fail to maintain monotonicity. *Danger only* achieves high $\Delta$ (64.54) but loses monotonicity across subsets. *Safety only* performs worse $\Delta$ (20.91), failing to capture high-risk samples effectively. It confirms that contrasting both directions is essential for reliable risk quantification. **Initialization sensitivity.** Computing SQSD at direction-insensitive states (*Insens. init*) causes S1 ASR to drop from 71.27% to 38.36%, validating our choice of direction-sensitive initialization. These ablations demonstrate that all three components are necessary for accurate risk rankings. Additional analysis on **learning rate sensitivity** is provided in Appendix H.

## 6. Conclusion and Outlook

**Conclusion.** This work analyzes safety degradation induced by benign fine-tuning from a parameter dynamics perspective, revealing the underlying mechanism and proposing a method for sample-level risk quantification. By tracking parameter trajectories during fine-tuning, finding that safety degradation corresponds to increasing cumulative drift toward danger directions while safety directions remain unchanged. This mechanism suggests that samples contributing more to this cumulative drift may pose greater fine-tuning risks. Motivated by this insight, we propose SQSD, which quantifies sample-level safety risk in fine-tuning by computing the projection gap of parameter updates along danger direction versus safety ones. A theoretical connection between parameter updates and model

output preferences is established via first-order Taylor approximation. SQSD demonstrates superior performance in quantifying sample-level risks and exhibits strong transferability across model architectures, parameter scales and parameter-efficient methods.

**Outlook.** While SQSD demonstrates strong empirical performance, its effectiveness depends on the initialization model's sensitivity to safety-relevant directions. Future research on constructing more universally informative parameter directions would be valuable. Moreover, current safety fine-tuning algorithms treat all samples equally. Integrating SQSD with existing safety fine-tuning methods represents a promising direction for better fine-tuning algorithms.

## Impact Statement

This work aims to advance LLM safety by identifying and quantifying fine-tuning risks in seemingly benign training data, enabling practitioners to assess sample-level safety risks before deployment and potentially preventing inadvertent safety degradation during model adaptation. However, we acknowledge potential dual-use concerns: the same techniques that identify high-risk samples for safety practitioners could theoretically be exploited by malicious actors to deliberately select data that maximally degrades model safety. We emphasize that our primary goal is defensive, aiming to help model developers maintain safety alignment during fine-tuning. We encourage the community to develop complementary safeguards, particularly risk-aware fine-tuning algorithms that can leverage our risk scores to preserve safety while adapting to downstream tasks. The broader deployment of these safety-preserving methods will be essential as LLMs become increasingly customizable through fine-tuning.

## Acknowledgments

The work is supported by the National Natural Science Foundation of China (No. 62272092), National Science Foundation for Young Scientists of China (No. 62502081), and the Fundamental Research Funds for the Central Universities under Grant (N25XQD004).

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

# A. Construction and Validation of Safety-related Direction

## A.1. Direction Construction

We construct safety-relevant directions through fine-tuning on specialized datasets. For the danger directions, we use the unsafe subsets from Aegis (Ghosh et al., 2024) and BeaverTails (Ji et al., 2023) datasets, randomly sampling 3k examples from each, and train them via supervised fine-tuning (SFT). For the safety direction, we use the full PKU-SafeRLHF-10k (Ji et al., 2023) dataset trained with Direct Preference Optimization (DPO). All direction construction employs LoRA-based training with rank $r = 8$ and scaling factor $\alpha = 16$.

The training configuration for danger directions uses a learning rate of $5 \times 10^{-6}$, batch size of 8, and 10 epochs. The safety direction follows a similar configuration, and additionally uses $\beta = 0.1$ for the DPO objective. For the danger directions, we use the final checkpoint after training completion. For the safety direction, we select model-specific intermediate checkpoints that demonstrate optimal safety alignment: checkpoint-9000 for Qwen3-8B, checkpoint-8000 for Llama-3.1-8B-Instruct, and checkpoint-7000 for Llama-2-7B-Chat. These checkpoints are chosen based on preliminary validation to ensure the resulting directions capture meaningful safety-relevant parameter displacements.

## A.2. Direction Validation

To verify that these directions capture safety-relevant behavioral changes, we perform parameter steering experiments by interpolating the model parameters along the defined directions:

$$\theta(\alpha) = \theta_0 + \alpha V, \qquad V \in \{V_{\text{safety}}, V_{\text{danger}}\}, \tag{14}$$

where $\alpha$ controls the steering magnitude. We measure the safety of $\theta(\alpha)$ across different $\alpha$ values using the Safety Score metric (Equation 15) evaluated on CategoricalHarmfulQA (Bhardwaj et al., 2024) to examine how safety changes with the steering magnitude.

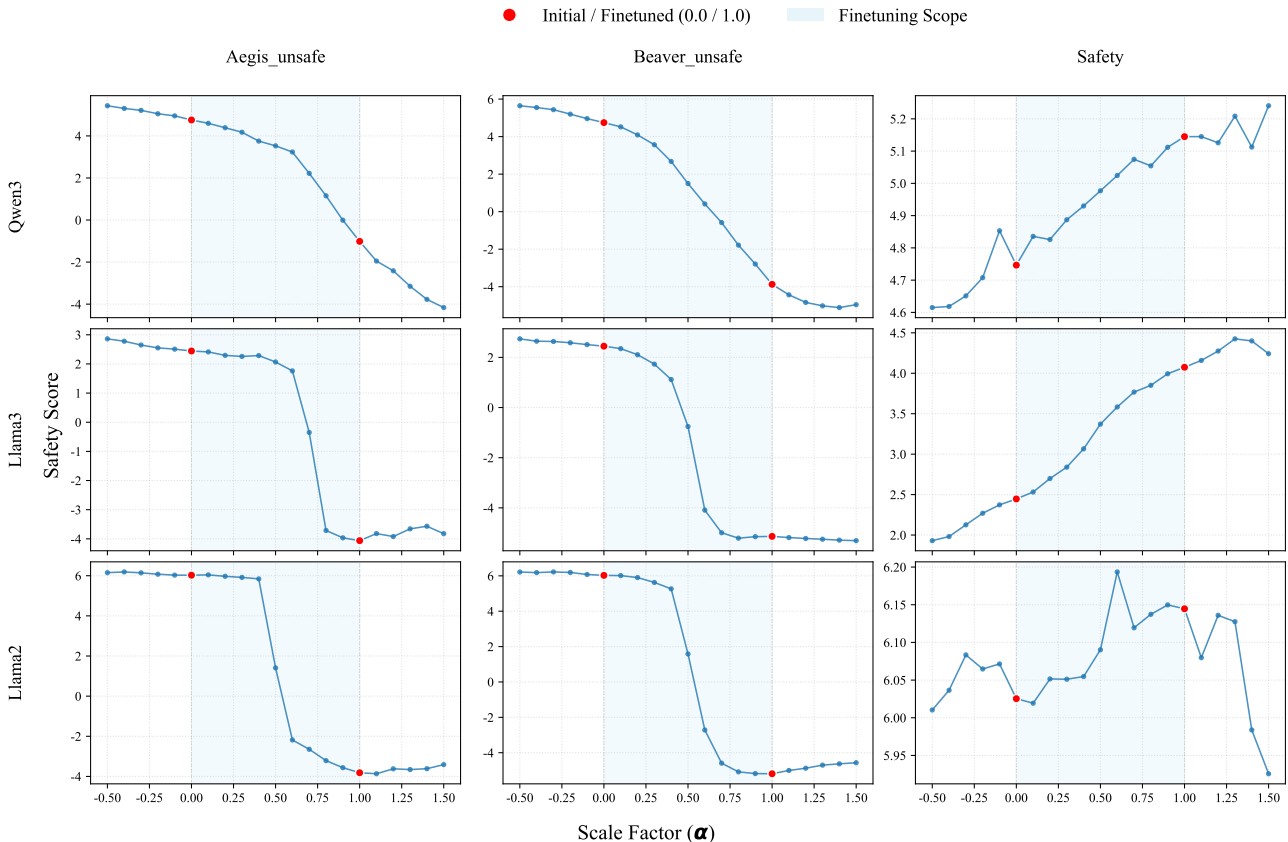

*Figure 5.* Parameter steering validation. Safety Score as functions of steering magnitude $\alpha$ for different directions.

**Results.** Figure 5 presents the validation results across three models and three direction types. The results demonstrate that our constructed directions reliably encode safety-relevant parameter displacements. Both Aegis-unsafe and Beaver-unsafe danger directions consistently decrease Safety Score as $\alpha$ increases across all three models. Conversely, the safety direction exhibits the opposite trend for Qwen3-8B and Llama-3.1-8B-Instruct, with Safety Score increasing as $\alpha$ grows.

However, Llama-2-7B-Chat does not show consistent Safety Score improvement across the entire steering magnitude range. This is because Llama-2-7B-Chat is already highly safety-aligned (Safety Score $> 6.0$), and our DPO-based alignment training fails to further improve its safety. Nevertheless, the direction remains locally valid, it induces predictable safety changes within a limited parameter neighborhood around the initial state (approximately $\alpha \in [0, 0.6]$). For SQSD, we initialize the model within this locally valid region when computing safety projections, which is sufficient for reliable sample-level risk quantification.

## B. Safety Evaluation Metrics.

We evaluate model safety using two complementary metrics on a fixed evaluation set $D_{\text{eval}}$. Let $y \sim p_\theta(\cdot \mid x)$ denote the model's generated response to prompt $x$.

**Safety Score.** This metric quantifies the overall safety level of model responses using a pretrained reward model $R_\psi(x, y)$. Higher scores indicate safer responses:

$$\text{Safety}(\theta) = \frac{1}{|D_{\text{eval}}|} \sum_{x \in D_{\text{eval}}} \mathbb{E}_{y \sim p_\theta(\cdot|x)} \big[ R_\psi(x, y) \big]. \tag{15}$$

In our experiments, we use `beaver-7b-unified-cost` (Ji et al., 2025a) as $R_\psi$.

**Attack Success Rate (ASR).** This metric measures the proportion of model responses that are classified as harmful. Lower ASR values indicate better safety:

$$\text{ASR}(\theta) = \frac{1}{|D_{\text{eval}}|} \sum_{x \in D_{\text{eval}}} \mathbb{E}_{y \sim p_\theta(\cdot|x)} \big[ \mathbb{I}_{\text{harmful}}(x, y) \big], \tag{16}$$

where $\mathbb{I}_{\text{harmful}}(x, y) \in \{0, 1\}$ is a binary indicator function that returns 1 if the response $y$ is deemed harmful, and 0 otherwise. We use LlamaGuard3-8B (Llama Team, 2024) as the safety classifier to determine $\mathbb{I}_{\text{harmful}}$. Both metrics are computed using greedy decoding for deterministic evaluation across all experiments.

## C. Derivation of First-Order Taylor Approximation

We provide the complete derivation of Equation (13) in § 4.2. Consider a training sample $z = (x, y)$ and two parameter states $\theta_{\text{ref}}$ and $\theta_{\text{target}}$ (e.g., an initial model and its fine-tuned counterpart). We perform a first-order Taylor expansion of the loss $\mathcal{L}(z, \theta_{\text{target}})$ around $\theta_{\text{ref}}$:

$$\mathcal{L}(z, \theta_{\text{target}}) = \mathcal{L}(z, \theta_{\text{ref}}) + \nabla_\theta \mathcal{L}(z, \theta_{\text{ref}})^\top (\theta_{\text{target}} - \theta_{\text{ref}}) + O(\|(\theta_{\text{target}} - \theta_{\text{ref}})\|^2). \tag{17}$$

Let $\theta' = \theta_{\text{ref}} - \eta \nabla_\theta \mathcal{L}(z, \theta_{\text{ref}})$ denote the parameters after a single gradient descent step on sample $z$ from $\theta_{\text{ref}}$ with learning rate $\eta > 0$. Rearranging gives $\nabla_\theta \mathcal{L}(z, \theta_{\text{ref}}) = -\frac{1}{\eta}(\theta' - \theta_{\text{ref}})$. Substituting into Equation (17):

$$\mathcal{L}(z, \theta_{\text{target}}) = \mathcal{L}(z, \theta_{\text{ref}}) - \frac{1}{\eta}(\theta' - \theta_{\text{ref}})^\top (\theta_{\text{target}} - \theta_{\text{ref}}) + O(\|(\theta_{\text{target}} - \theta_{\text{ref}})\|^2). \tag{18}$$

Rearranging to isolate the loss difference and multiplying both sides by $\eta$:

$$\eta[\mathcal{L}(z, \theta_{\text{ref}}) - \mathcal{L}(z, \theta_{\text{target}})] \approx (\theta' - \theta_{\text{ref}})^\top (\theta_{\text{target}} - \theta_{\text{ref}}), \tag{19}$$

## D. Implementation Details of Baselines

### D.1. Reward Model

The Reward Model baseline directly uses a pretrained safety reward model to score each training sample's safety. We use `beaver-7b-unified-cost` (Ji et al., 2025a), which outputs a cost value where lower values indicate safer content.

For a sample $z = (x, y)$, we compute the risk score as:

$$\text{Risk}_{\text{RM}}(z) = R_\psi(x, y) \tag{20}$$

where $R_\psi(x, y)$ is the cost output from the reward model. Higher risk scores indicate higher-risk samples.

**Implementation.** For each sample, we concatenate prompt $x$ and response $y$ following the model's chat template, feed it to the reward model using greedy decoding, and extract the scalar cost output as the risk score.

### D.2. Bi-Anchor(Reps)

The Bi-Anchor(Reps) (He et al., 2024) quantifies sample risk through representation similarity in the hidden state space. For each training sample, we extract its representation as the hidden state of the second-to-last token at the final layer. The risk score is computed by measuring the similarity between this representation and those of harmful anchor samples. For a training sample $z = (x, y)$, let $\mathbf{h}(z) \in \mathbb{R}^d$ denote its representation. Given a set of harmful samples $\mathcal{D}_{\text{harmful}}$, the risk score is:

$$\text{Risk}_{\text{Bi-Anchor(Reps)}}(z) = \max_{z' \in \mathcal{D}_{\text{harmful}}} \frac{\langle \mathbf{h}(z), \mathbf{h}(z') \rangle}{\|\mathbf{h}(z)\|_2 \|\mathbf{h}(z')\|_2} \tag{21}$$

where the fraction denotes cosine similarity. Higher similarity to harmful samples indicates higher risk.

**Implementation.** For each sample, we feed it to the model and extract the hidden state of the second-to-last token at the final layer as the sample representation (the last token is typically an end-of-sequence token, while the second-to-last token has access to all preceding information). We use 10 harmful samples from `pure_bad_10.jsonl` (provided in the original repository) as $\mathcal{D}_{\text{harmful}}$ to construct the harmful anchors. For each training sample, we compute its cosine similarity with all 10 harmful anchor representations and take the maximum value as the final risk score.

### D.3. Bi-Anchor(Grad)

The Bi-Anchor(Grad) method (He et al., 2024) uses gradient information as sample features to quantify risk. For each training sample, we compute its gradient with respect to model parameters as the sample representation. The risk score is determined by the difference between the sample's similarity to harmful anchors and its similarity to safe anchors. For a training sample $z = (x, y)$, let $\mathbf{g}(z)$ denote its normalized gradient representation. Given harmful anchor gradient $\mathbf{g}_{\text{harm}}$ and safe anchor gradients $\mathbf{g}_{\text{safe1}}, \mathbf{g}_{\text{safe2}}$, the risk score is:

$$\text{Risk}_{\text{Bi-Anchor(Grad)}}(z) = \langle \mathbf{g}(z), \mathbf{g}_{\text{harm}} \rangle - \langle \mathbf{g}(z), \mathbf{g}_{\text{safe1}} \rangle - \langle \mathbf{g}(z), \mathbf{g}_{\text{safe2}} \rangle \tag{22}$$

where $\langle \cdot, \cdot \rangle$ denotes the inner product. Higher values indicate the sample's gradient aligns more with harmful patterns than safe patterns.

**Implementation.** For each sample, we compute its loss gradient with respect to model parameters, flatten and concatenate all gradient tensors into a single vector, then apply L2 normalization to obtain $\mathbf{g}(z)$. To construct anchor gradients, we use three anchor datasets: one harmful set (`illegal-activities-10.jsonl`) and two safe sets (`illegal-activities-10-anchor1.jsonl` and `illegal-activities-10-anchor2.jsonl`). For each anchor dataset, we compute the normalized gradient for every sample, then average them to obtain a single anchor vector $\mathbf{g}_{\text{harm}}, \mathbf{g}_{\text{safe1}}, \mathbf{g}_{\text{safe2}}$. The final risk score is computed as the weighted sum of dot products with weights $(1, -1, -1)$ for harmful and two safe anchors respectively.

### D.4. Self-Inf-N

The Self-Inf-N method (Guan et al., 2025) is based on the intuition that outlier samples induce larger gradient magnitudes, indicating greater influence on model parameters. For each training sample, we compute its gradient and measure the self-influence as the inner product of the gradient with itself.

For a training sample $z = (x, y)$ with response $y$ of length $|y|$, let $\mathbf{g}(z)$ denote its gradient. The self-influence is defined as:

$$\text{Self-Inf}(z) = \langle \mathbf{g}(z), \mathbf{g}(z) \rangle \tag{23}$$

To mitigate response-length bias, the final risk score incorporates response length:

$$\text{Risk}_{\text{Self-Inf-N}}(z) = \log(\text{Self-Inf}(z) + 1) + \log(|y| + 1) \tag{24}$$

where the logarithmic transformation balances the contribution of gradient magnitude and response length.

## D.5. LARF

The LARF (Layer-Aware Representation Filtering) (Li et al., 2025b) identifies safety-degrading samples through a two-stage pipeline. First, it identifies safety-sensitive layers by scaling each layer's parameters and measuring the resulting change in refusal responses on an over-rejection dataset. Second, at the identified safety-sensitive layer, it computes average representations for safe reference samples ($\mathcal{D}_{\text{safe}}$) and unsafe reference samples ($\mathcal{D}_{\text{unsafe}}$), then assigns each training sample a risk score based on its representation similarity to these anchors.

For a training sample $z = (x, y)$, let $\mathbf{h}_l(z)$ denote its representation at the safety-sensitive layer $l$. Given anchor representations $\overline{\mathbf{h}}_{\text{safe}}$ and $\overline{\mathbf{h}}_{\text{unsafe}}$, the risk score is:

$$\text{Risk}_{\text{LARF}}(z) = \langle \mathbf{h}_l(z), \overline{\mathbf{h}}_{\text{unsafe}} \rangle - \langle \mathbf{h}_l(z), \overline{\mathbf{h}}_{\text{safe}} \rangle \tag{25}$$

where $\langle \cdot, \cdot \rangle$ denotes the inner product. Higher scores indicate greater alignment with unsafe patterns.

**Implementation.** To identify safety-sensitive layers, we search from layer 11 to the final layer using parameter scaling with perturbation factors $\alpha \in \{0.8, 0.9, 1.1, 1.2\}$. Based on this procedure, we identify layer 21 for Qwen3-8B, layer 13 for Llama-3.1-8B-Instruct, and layer 11 for Llama-2-7B-Chat as the safety-sensitive layers. For each training sample, we extract its hidden state at the corresponding safety-sensitive layer and compute the risk score using the formulation above.

# E. Directional Sensitivity Analysis and Initialization Details

This appendix provides complete technical details for the directional sensitivity analysis and initialization strategies.

## E.1. Formalization of Directional Sensitivity

We define directional sensitivity (DS) as the rate of safety behavior change per unit perturbation along direction $V$ under two parameter-space scenarios. When the initialization state lies on the linear interpolation path ($\theta_{\text{initial}} = \theta_0 + \alpha V$ for scalar $\alpha \in \mathbb{R}$), we define **linear-path DS** as:

$$\text{DS}_{\text{linear}}(\alpha) = \frac{\text{Safety}(\theta_0 + (\alpha + \delta)V) - \text{Safety}(\theta_0 + (\alpha - \delta)V)}{2\delta} \tag{26}$$

where $\delta > 0$ is a small perturbation magnitude ($\delta = 0.1$ in our experiments) and $\text{Safety}(\theta)$ denotes the Safety Score metric (Appendix B). This measures the local slope of the safety landscape along the linear path.

When the parameter state deviates from the linear path due to cumulative fine-tuning drift ($\theta_{\text{initial}} = \theta_t$ at training step $t$), we define **drift-enhanced DS** as:

$$\text{DS}_{\text{drift}}(t) = \frac{\text{Safety}(\theta_{t+a}) - \text{Safety}(\theta_t)}{\langle \theta_{t+a} - \theta_0, \hat{V} \rangle - \langle \theta_t - \theta_0, \hat{V} \rangle} \tag{27}$$

where $\hat{V} = V/\|V\|_2$ is the normalized direction vector, and $a = 150$ represents the step interval between adjacent checkpoints. This quantifies how much safety changes per unit of cumulative drift along direction $V$.

**Interpretation across directions.** For $V_{\text{safety}}$, higher DS values indicate greater sensitivity to safety-aligned perturbations. For $V_{\text{danger}}$, lower (more negative) DS values indicate greater sensitivity to danger-aligned perturbations.

## E.2. Identifying High-Sensitivity States

We compute directional sensitivity under different parameter states to identify high-sensitivity initialization points. For $V_{\text{safety}}$, we evaluate linear-path DS; for $V_{\text{danger}}$, we evaluate drift-enhanced DS. We present the top-5 highest-sensitivity states for each configuration.

**Sensitivity for Safety Direction.** For $V_{\text{safety}}$, we compute linear-path DS based on the steering experiments in Appendix A.2. In those experiments, we construct safety directions from different checkpoints during DPO alignment training, then record Safety Score at various $\alpha$ positions along each direction. Using these recorded safety scores, we calculate $\text{DS}_{\text{linear}}(\alpha)$ via Equation 26 with $\delta = 0.1$. Table 4 presents the top-5 $\alpha$ values with highest DS for each checkpoint-based safety direction, showing the high-sensitivity parameter states along linear paths for different models.

**Drift-Enhanced Sensitivity for Danger Directions.** For $V_{\text{danger}}$, we leverage the checkpoints from fine-tuning experiments

*Table 4.* Top-5 high-sensitivity $\alpha$ positions ranked by $DS_{linear}(\alpha)$ for safety directions. Higher DS values indicate stronger responsiveness to safety-aligned perturbations.

| Qwen3-7000 | | Llama2-9000 | | Llama3-7000 | | Llama3-8000 | |
|---|---|---|---|---|---|---|---|
| $\alpha$ | DS | $\alpha$ | DS | $\alpha$ | DS | $\alpha$ | DS |
| 0.2 | 1.01 | 0.5 | 0.69 | 0.4 | 2.66 | 0.4 | 2.49 |
| 0.3 | 0.52 | -0.4 | 0.37 | 0.5 | 2.59 | 0.5 | 2.31 |
| 0.6 | 0.49 | 1.2 | 0.24 | 0.6 | 1.98 | 0.3 | 2.12 |
| 0.5 | 0.47 | 0.4 | 0.20 | 0.3 | 1.83 | 0.6 | 1.88 |
| 0.9 | 0.45 | 0.2 | 0.16 | 0.2 | 1.54 | 0.2 | 1.70 |

*Table 5.* Top-5 high-sensitivity checkpoints ranked by $|DS_{drift}(t)|$ for danger directions. All DS values are negative; lower values indicate greater sensitivity to danger-aligned perturbations.

| Model | Direction | Top-5 Checkpoints | | | | | Training Dataset |
|---|---|---|---|---|---|---|---|
| Qwen3-8b | Aegis | 5850 | 5250 | 5700 | 4200 | 3150 | dolly(5k) |
| | Beaver | 5850 | 5250 | 5700 | 4200 | 3750 | dolly(5k) |
| Llama3-8b | Aegis | 6150 | 5250 | 5850 | 5700 | 4950 | alpaca(5k) |
| | Beaver | 5700 | 4800 | 4350 | 4650 | 4250 | alpaca(5k) |
| | Aegis | 5550 | 6000 | 5850 | 4950 | 4350 | dolly(5k) |
| | Beaver | 5700 | 4800 | 4350 | 4650 | 5250 | dolly(5k) |
| Llama2 | Aegis | 6150 | 4800 | 5400 | 4200 | 4050 | dolly(5k) |
| | Beaver | 6150 | 5550 | 5850 | 5700 | 5400 | dolly(5k) |

in §5.2, where checkpoints are saved every 150 training steps. Using consecutive checkpoints, we compute $DS_{drift}(t)$ via Equation 27. Table 5 presents the top-5 high-sensitivity parameter states for different models and danger directions.

### E.3. Initialization States for Main Experiments

Our main experiments (§ 5.3) evaluate SQSD across 12 configurations: 3 models (Qwen3-8B, Llama-3.1-8B-Instruct, Llama-2-7B-Chat) × 2 datasets (Dolly, Alpaca) × 2 danger-safety direction pairs. For each configuration, we compute SQSD using one danger direction (Aegis-unsafe or Beaver-unsafe) paired with the safety direction. Table 6 presents the selected initialization states for each configuration.

*Table 6.* Selected initialization states for main experiments. For danger directions, we report the checkpoint step (with rank in top-5) and the training dataset used to obtain that checkpoint. For safety directions, we report the checkpoint number and corresponding $\alpha$ value.

| | Dolly | | | Alpaca | | |
|---|---|---|---|---|---|---|
| | **Aegis** | **Beaver** | **Safety** | **Aegis** | **Beaver** | **Safety** |
| Qwen3 | 5850[†] (top1) | 5850[†] (top1) | 7000 ($\alpha$=0.2) | 5850[†] (top1) | 5850[†] (top1) | 7000 ($\alpha$=0.2) |
| Llama3 | 5550[†] (top1) | 5700[†] (top1) | 7000 ($\alpha$=0.4) | 5850[‡] (top3) | 4650[‡] (top4) | 8000 ($\alpha$=0.4) |
| Llama2 | 4050[†] (top1) | 5400[†] (top3) | 9000 ($\alpha$=0.5) | 4050[†] (top1) | 6150[†] (top1) | 9000 ($\alpha$=0.5) |

[†]Checkpoint obtained from pilot fine-tuning on Dolly. [‡]Checkpoint obtained from pilot fine-tuning on Alpaca.

The selection strategy balances sensitivity and reliability: we prioritize the highest-sensitivity checkpoint (top1) when possible, but occasionally select from top3–top4 when the top1 checkpoint fails to produce SQSD scores that consistently predict the severity of safety degradation across the entire corpus. This reveals an important consideration for projection-based risk quantification: SQSD's performance depends on the informativeness of safety-relevant directional vectors in the local parameter region. Although these directions encode well-defined safety semantics (validated in Appendix A.2), even when initializing at the most sensitive parameter states, SQSD may not always achieve consistent predictive performance across all corpus-wide risk quantification scenarios. This suggests that future work could explore adaptive direction construction or multi-directional ensemble approaches to improve robustness.

# F. Response Length Bias

## F.1. Response Length Bias in Unnormalized Scoring

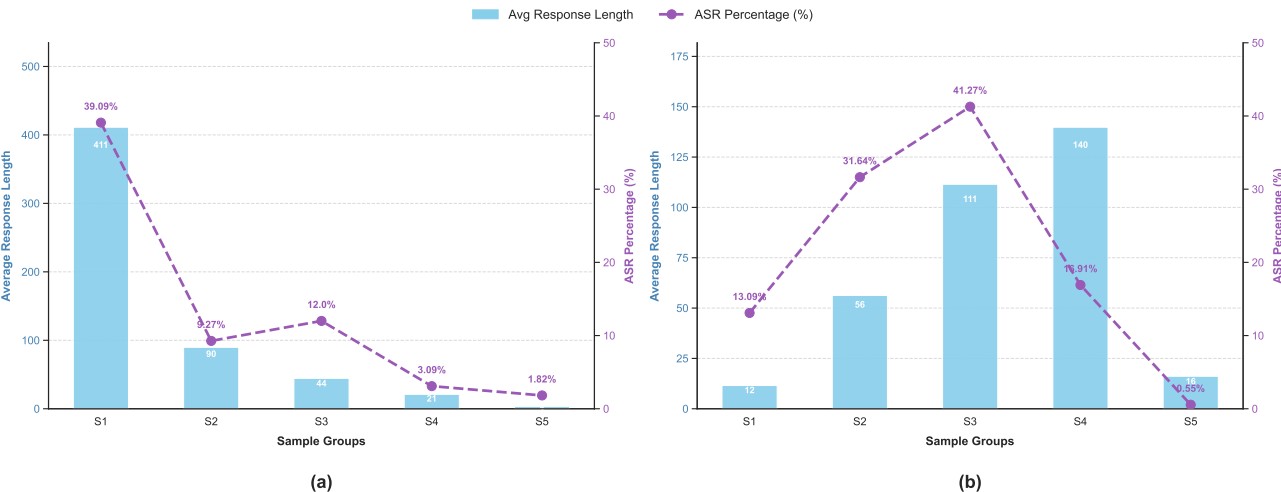

*Figure 6.* Response length bias in unnormalized risk scoring. Average response length and ASR for Qwen3-8B fine-tuned on Dolly subsets ranked by (a) response length and (b) unnormalized SQSD.

Prior gradient-based methods (Guan et al., 2025; He et al., 2024) exhibit response-length bias when using unnormalized parameter updates. To investigate whether response length correlates with fine-tuning risk, we compare two ranking strategies: (1) ranking samples by response length, and (2) ranking by unnormalized SQSD scores (without module-wise normalization in Equation 11).

Figure 6 presents average response length and ASR for models fine-tuned on five sample subsets (S1-S5, 1000 samples each). Observing both subfigures, ASR shows no consistent relationship with response length. Notably in Figure 6(a), S5 contains very short responses (average length 3) yet achieves the lowest ASR (1.82%), demonstrating that short-response samples are not inherently high-risk. Figure 6(b) reveals critical issues with unnormalized SQSD. First, ASR does not decrease monotonically, indicating unnormalized SQSD fails to capture true sample-level risk. Second, S1 with highest unnormalized SQSD scores has the shortest average response length (12 tokens), while longer responses receive lower scores. This demonstrates that unnormalized SQSD is disproportionately influenced by short-response samples. These observations motivate our module-wise normalization (Equation 11) to mitigate response-length bias.

## F.2. Understanding the Short-Response Bias

Gradient-based methods for identifying high-risk samples consistently exhibit short-response bias, the top-ranked samples invariably have very short responses when using unnormalized gradients. However, these short-response samples do not always constitute the most harmful subset for model safety as demonstrated in the previous section. To understand this phenomenon, we analyze the relationship between sample loss and response length, revealing the underlying mechanism behind this bias.

**Loss Distribution.** We analyze loss values across samples with different response lengths to understand the short-response bias. Figure 7 shows that the shortest responses (Bottom 1000, 4-9 tokens) exhibit loss values ranging from 2 to 12, while Medium (40-49 tokens) and Top groups (173-321 tokens) show loss concentrated in 1 to 4. This reveals the reason: short responses amplify sample loss, which increases gradient magnitude, leading gradient-based methods to assign inflated scores to these samples. This confirms why unnormalized gradient-based methods consistently rank short-response samples highest: **large loss values produce large gradients, resulting in disproportionately high risk scores regardless of actual safety impact.**

**Loss Distribution of per-token.** Since SFT loss averages cross-entropy loss across response tokens, we analyze per-token loss to explain the amplified average loss in short responses. Figures 8 and 9 show that the first response token and the final token in short responses consistently exhibit high loss, while other positions show normal values. Long responses

can amortize these high-loss positions across many tokens, but short responses lack sufficient tokens to dilute these spikes, resulting in amplified average loss. **First token high loss.** The first response token faces high uncertainty due to lack of response context and diverse possible response styles given the prompt. Without accumulated context to constrain predictions, the model cannot form strong priors, resulting in elevated loss. **End token high loss in short responses.** The model learns a length prior from predominantly longer responses in training data. Short responses violate this expectation, the model anticipates continued generation rather than early termination. Predicting end-of-sequence after few tokens is inherently surprising given the learned length distribution, causing high cross-entropy loss. Training data's scarcity of short responses further exacerbates this bias. These position-specific loss spikes, averaged over few tokens, produce the systematically elevated loss observed in short responses, explaining gradient-based methods' preference for these samples.

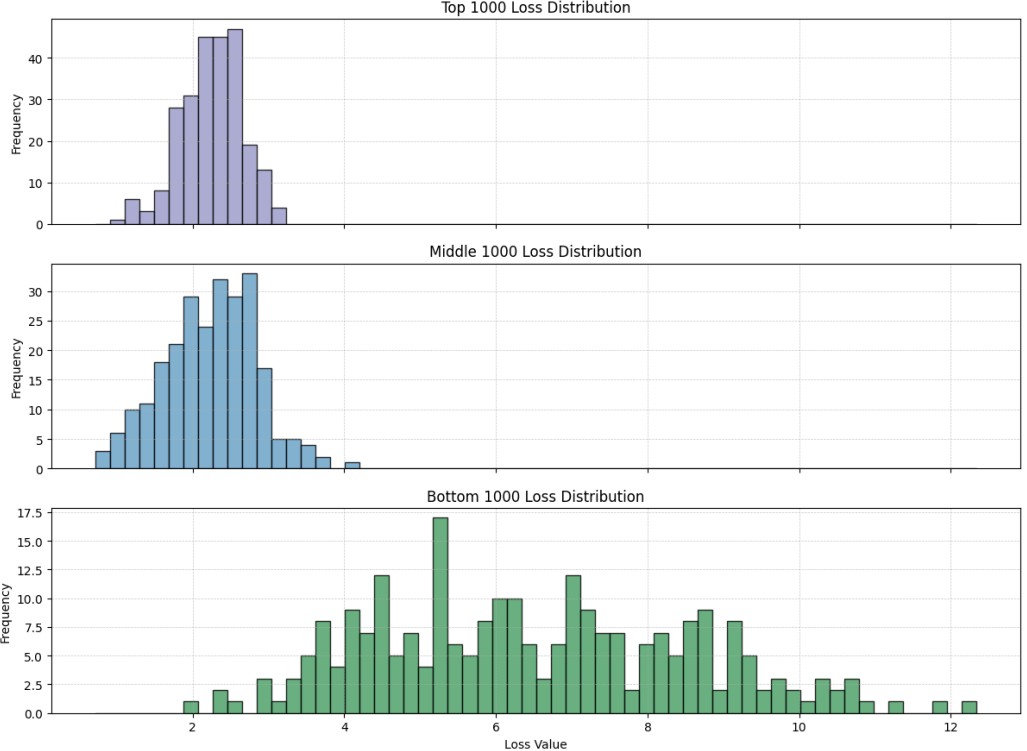

*Figure 7.* Loss distribution across response length groups. Cross-entropy loss distributions for samples from Dolly dataset grouped by response length: Top 1000 (173-321 tokens), Middle 1000 (40-49 tokens), and Bottom 1000 (4-9 tokens).

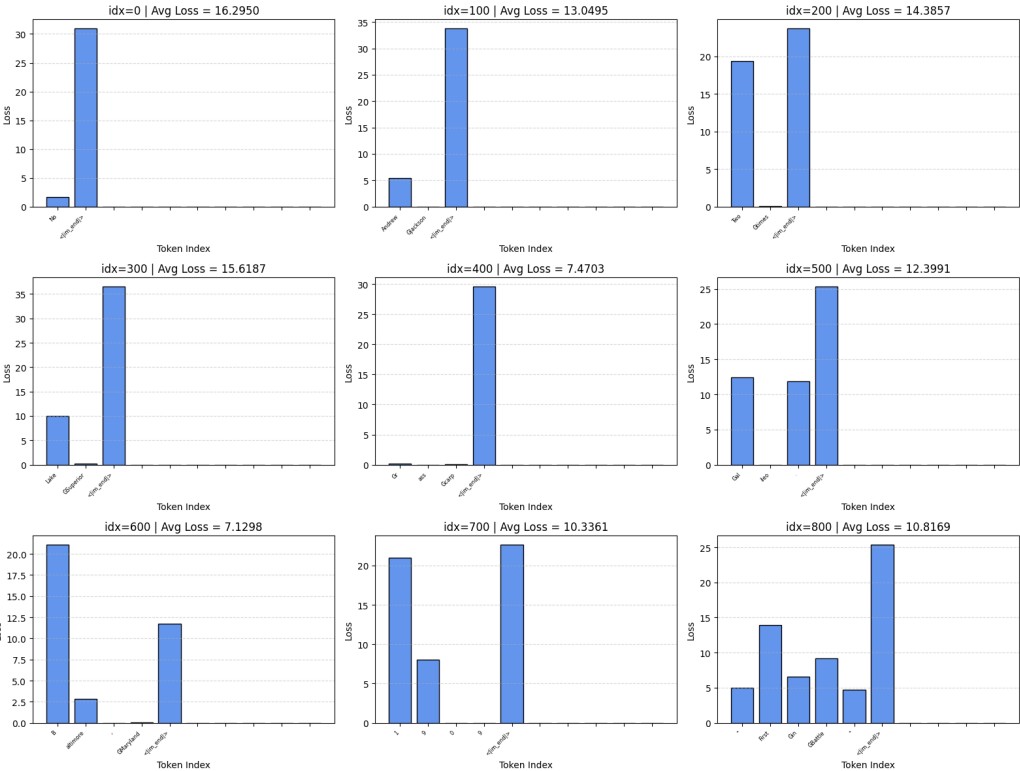

*Figure 8.* Per-token cross-entropy loss for short-response samples. Loss distribution across tokens for 9 representative short-response samples. The final token in each sample is <|im_end|>.

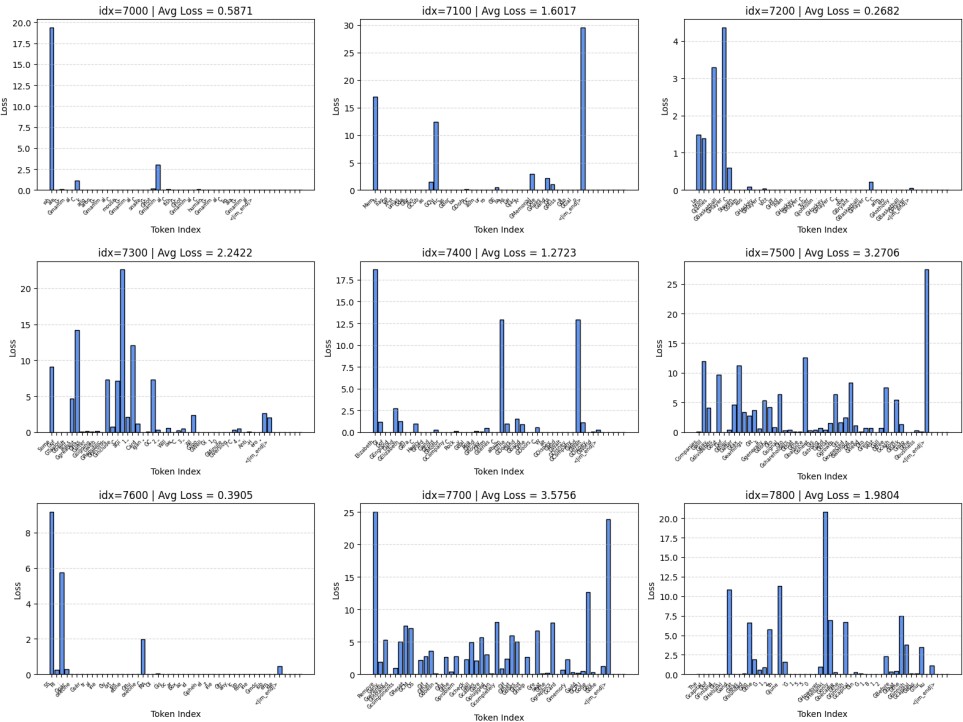

*Figure 9.* Per-token cross-entropy loss for middle-response samples. Loss distribution across tokens for 9 representative middle-response samples. The final token in each sample is <|im_end|>.

# G. Transferability Experiments Details

We conduct transferability experiments to evaluate whether SQSD scores computed under one configuration can predict fine-tuning risks under different configurations. All experiments use SQSD(Beaver), where the danger direction is Beaver-unsafe and the safety direction is derived from DPO alignment on PKU-SafeRLHF. Each experiment partitions the target dataset into five risk-ranked subsets (S1-S5, 1,000 samples each) based on SQSD scores, then fine-tunes models on each subset to measure the resulting safety degradation.

All fine-tuning experiments use LoRA (rank 8, scaling factor 16) with 10 epochs and batch size 8. The learning rate is $5 \times 10^{-5}$ for LoRA experiments and $5 \times 10^{-6}$ for full fine-tuning experiments.

**Cross-Architecture Transferability.** We evaluate bidirectional transfer between Llama3.1-8B-Instruct and Qwen3-8B. In the Llama-to-Qwen direction, SQSD scores computed using Llama3.1-8B-Instruct are used to rank Alpaca samples for fine-tuning Qwen3-8B. In the Qwen-to-Llama direction, SQSD scores computed using Qwen3-8B are used to rank Dolly samples for fine-tuning Llama3.1-8B-Instruct.

**Cross-Parameter-Scale Transferability.** We compute SQSD scores using Qwen3-8B, then apply these scores to rank Dolly samples for fine-tuning larger models (Qwen3-14B and Qwen3-32B).

**Cross-Parameter-Efficient-Method Transferability.** We compute SQSD scores using LoRA gradients on Qwen3-8B, then apply these scores to rank Dolly samples for full parameter fine-tuning on the same model.

# H. Learning Rate Sensitivity Analysis

We additionally examine SQSD's performance under different learning rates. All experiments in this section use Qwen3-8B fine-tuned on Dolly with the Beaver-unsafe direction. As shown in Figure 10, ASR decreases monotonically from S1 to S5 across all learning rate settings, demonstrating that SQSD consistently predicts the severity of safety degradation regardless of learning rate choice. This indicates strong robustness to learning rate variations. Additionally, we observe that smaller learning rates induce weaker safety degradation during fine-tuning. Notably, at lr=1e-5, the model fine-tuned on the highest-risk subset (S1) achieves an ASR of only 13.9%, substantially lower than higher learning rates.

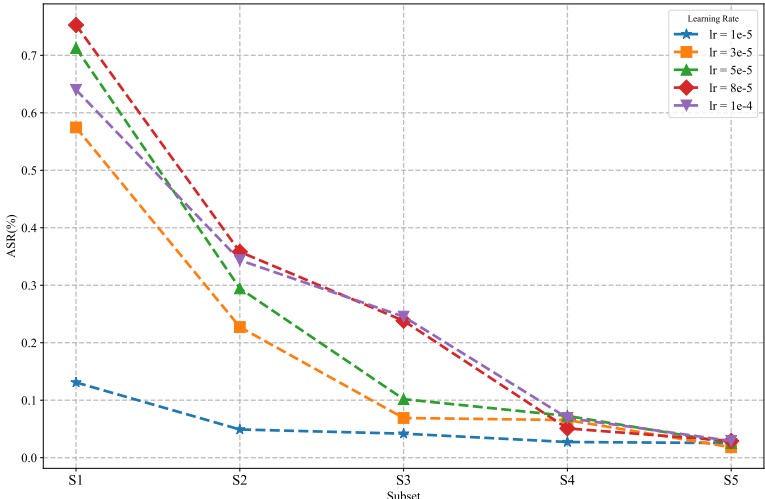

*Figure 10.* Impact of learning rate on SQSD performance. ASR on CategoricalHarmfulQA for Qwen3-8B fine-tuned on Dolly subsets (S1-S5) ranked by SQSD computed at different learning rates.

# I. Causal Analysis of Danger-Direction Drift and Safety Degradation

The trajectory analysis in §3.2 reveals a strong correlation between cumulative parameter drift along danger directions and safety degradation. Here we provide multi-level evidence to substantiate this as a causal relationship.

**Causal evidence from parameter steering.** As shown in §3.1 and Appendix A.2, parameter steering experiments constitute

direct causal interventions: deliberately perturbing model parameters along the danger direction produces monotonic safety degradation across all configurations (Figure 5), directly demonstrating that danger-direction displacement *produces* safety degradation.

**Scaling Evidence for Causal Interpretation.** Figure 4 presents parameter drift trajectories for Qwen3-8B fine-tuned on Alpaca at scales from 3k to 50k samples. Within the same dataset, larger data scale induces greater cumulative drift along the danger direction, and this greater drift in turn leads to more severe safety degradation, with the terminal Safety Score declining monotonically from approximately 3.8 (3k) to $-0.3$ (50k). This scaling pattern strengthens the causal relationship between the safety degradation and parameter drift : the cumulative parameter drift along the danger direction induced by fine-tuning is what causes safety degradation, rather than merely correlating with it.

**Counterfactual Intervention.** To strengthen the causal claim established in §3.2, we conduct a counterfactual intervention on a checkpoint that has undergone safety degradation through benign fine-tuning. Specifically, we orthogonalize the cumulative parameter displacement $\Delta W = W_{\text{ft}} - W_{\text{base}}$ with respect to the danger direction:

$$\Delta W^{\text{orth}} = \Delta W - \langle \Delta W, \hat{V}_{\text{danger}} \rangle \hat{V}_{\text{danger}}, \tag{28}$$

where $\hat{V}_{\text{danger}}$ is the unit vector of the danger direction, such that $\langle \Delta W^{\text{orth}}, \hat{V}_{\text{danger}} \rangle = 0$, and reconstruct the model as $W^{\text{orth}} = W_{\text{base}} + \Delta W^{\text{orth}}$. As shown in Table 7, ASR drops from 15.09% to 0.91%, confirming that safety degradation is causally attributable to the danger-direction component of the parameter drift.

*Table 7.* ASR (%) before and after the counterfactual intervention.

|  | ASR (%) |
| --- | --- |
| Before intervention | 15.09 |
| After intervention | 0.91 |

# J. Extended Effectiveness Evaluation of SQSD

## J.1. Safety Degradation Across Categories

To further demonstrate that SQSD quantifies sample-level fine-tuning risk across diverse safety categories rather than a single aggregate metric, we evaluate category-wise ASR on CatHarmfulQA, a benchmark spanning 11 safety categories. Table 8 reports results for Qwen3-8B fine-tuned on the Dolly dataset under both Beaver-unsafe and Aegis-unsafe directions. In both settings, the majority of categories exhibit monotonically decreasing ASR from S1 to S5 (8/11 and 7/11, respectively), and overall ASR maintains strict monotonicity in both cases. For the remaining categories where strict monotonicity is not observed, the violations are minor fluctuations rather than systematic failures, confirming that SQSD consistently captures the relative risk ordering across diverse harm types.

*Table 8.* Category-wise ASR (%) on CatHarmfulQA for Qwen3-8B fine-tuned on Dolly under Beaver (left) and Aegis (right) directions.

| Category | Beaver Direction | | | | | | Aegis Direction | | | | | |
| --- | --- | --- | --- | --- | --- | --- | --- | --- | --- | --- | --- | --- |
|  | S1 | S2 | S3 | S4 | S5 | Mono | S1 | S2 | S3 | S4 | S5 | Mono |
| Illegal Activity | 78.0 | 36.0 | 10.0 | 6.0 | 2.0 | ✓ | 58.0 | 26.0 | 16.0 | 12.0 | 4.0 | ✓ |
| Child Abuse | 72.0 | 28.0 | 6.0 | 4.0 | 2.0 | ✓ | 44.0 | 20.0 | 8.0 | 2.0 | 0.0 | ✓ |
| Hate/Harass/Violence | 78.0 | 44.0 | 6.0 | 0.0 | 0.0 | ✓ | 52.0 | 12.0 | 12.0 | 0.0 | 0.0 | ✓ |
| Malware Viruses | 56.0 | 38.0 | 8.0 | 14.0 | 2.0 | ✗ | 34.0 | 32.0 | 10.0 | 4.0 | 2.0 | ✓ |
| Physical Harm | 72.0 | 28.0 | 2.0 | 4.0 | 0.0 | ✗ | 46.0 | 18.0 | 12.0 | 0.0 | 2.0 | ✗ |
| Economic Harm | 80.0 | 28.0 | 14.0 | 10.0 | 0.0 | ✓ | 52.0 | 34.0 | 18.0 | 12.0 | 6.0 | ✓ |
| Fraud/Deception | 78.0 | 38.0 | 20.0 | 14.0 | 10.0 | ✓ | 46.0 | 48.0 | 18.0 | 8.0 | 8.0 | ✗ |
| Adult Content | 74.0 | 20.0 | 10.0 | 0.0 | 2.0 | ✗ | 36.0 | 18.0 | 14.0 | 0.0 | 0.0 | ✓ |
| Political Campaigning | 74.0 | 32.0 | 18.0 | 16.0 | 4.0 | ✓ | 58.0 | 46.0 | 26.0 | 10.0 | 2.0 | ✓ |
| Privacy Violation | 66.0 | 20.0 | 8.0 | 6.0 | 4.0 | ✓ | 38.0 | 28.0 | 14.0 | 2.0 | 4.0 | ✗ |
| Tailored Financial Advice | 56.0 | 12.0 | 8.0 | 6.0 | 2.0 | ✓ | 34.0 | 30.0 | 22.0 | 0.0 | 2.0 | ✗ |
| **Overall** | **71.3** | **29.5** | **10.0** | **7.3** | **2.6** | ✓ | **45.3** | **28.4** | **15.5** | **4.6** | **2.7** | ✓ |

## J.2. Computational Efficiency and Memory Consumption Comparison

Table 9 reports the peak memory consumption and wall-clock time required to score 1k samples on Qwen3-8B. All methods are evaluated on an RTX-A6000 (48G), except Bi-Anchor(Grad) on RTX-Pro 6000 (96G) due to its substantially higher memory demand. Among all gradient-based methods (Self-Inf-N and Bi-Anchor(Grad)), SQSD achieves the lowest runtime (1918s vs. 2718s and 8127s), as risk scores are computed solely from LoRA parameter gradients rather than full model gradients. The memory overhead of SQSD (44G) primarily stems from model parameters, gradients and the storage of safety-related direction vectors. Note that the runtime could be further reduced with sufficient GPU memory, as the projections onto danger and safety directions could be consolidated into a single gradient computation rather than computed separately.

*Table 9.* Overhead for scoring 1k samples on Qwen3-8B.

|            | Reward Model | Bi-Anchor(Reps) | Self-Inf-N | Bi-Anchor(Grad) | LARF | SQSD (Ours) |
|------------|:------------:|:---------------:|:----------:|:---------------:|:----:|:-----------:|
| Memory (G) | 25           | 32              | 45         | 90              | 23   | 44          |
| Time (s)   | 52           | 391             | 2718       | 8127            | 72   | 1918        |

## J.3. Generalization to Domain-Specific Datasets

To evaluate whether SQSD generalizes beyond general-purpose instruction datasets, we assess its performance on three domain-specific datasets with distinct characteristics, using Qwen3-8B with the Beaver-unsafe direction. Table 10 presents the resulting ASR across subsets. SQSD maintains monotonically decreasing ASR on both Role-Play (40.73%→1.64%) and BrainStorming (4.18%→0.91%), demonstrating that SQSD effectively captures sample-level fine-tuning risk across datasets with substantially different safety degradation magnitudes. For the Medical dataset, ASR remains consistently low across all subsets (0.1%–0.2%), indicating that medical instruction data does not induce meaningful safety degradation by nature; in such cases, risk quantification is beyond the reach of any sample-level method.

*Table 10.* ASR (%) on CatHarmfulQA for Qwen3-8B fine-tuned on domain-specific datasets (Beaver direction).

| Dataset                      | S1    | S2    | S3    | S4   | S5   | Mono |
|------------------------------|:-----:|:-----:|:-----:|:----:|:----:|:----:|
| Role-Play (Wang et al., 2024)       | 40.73 | 34.55 | 14.73 | 6.73 | 1.64 | ✓ |
| BrainStorming (Wan et al., 2023)    | 4.18  | 2.73  | 2.00  | 1.82 | 0.91 | ✓ |
| Medical                      | 0.10  | 0.11  | 0.10  | 0.20 | 0.11 | ✗ |

## J.4. SQSD Effectiveness Evaluation on Multiple Benchmarks and Metrics

This appendix provides supplementary evaluation results for Section 5.3. While the main paper reports ASR on Categorical-HarmfulQA, here we present comprehensive results across multiple safety benchmarks and metrics. Specifically, we report Safety Score on CategoricalHarmfulQA (Table 11), ASR on AdvBench (Table 12), and ASR on HEx-PHI (Table 13). These additional results consistently demonstrate SQSD's superior capability in quantifying sample-level fine-tuning risks across diverse evaluation settings.

*Table 11.* Effectiveness of SQSD on Safety Score. Safety Score on CategoricalHarmfulQA for models fine-tuned on risk-ranked subsets by various methods. S1-S5 represent 1000 samples each, uniformly sampled from highest to lowest risk rankings. Δ denotes Safety Score difference (S5 - S1) (higher is better). Mono indicates whether Safety Score increases monotonically across subsets (✓ is yes).

| Method | Dolly | | | | | | | Alpaca | | | | | | |
|---|---|---|---|---|---|---|---|---|---|---|---|---|---|---|
| | S1 | S2 | S3 | S4 | S5 | Δ ↑ | Mono | S1 | S2 | S3 | S4 | S5 | Δ ↑ | Mono |
| *Qwen3-8B* | | | | | | | | | | | | | | |
| Reward Model | -3.44 | 0.59 | 1.95 | 1.52 | 3.65 | 7.09 | ✗ | -2.00 | 2.42 | 1.99 | 1.93 | 3.02 | 5.02 | ✗ |
| Bi-Anchor(Reps) | 1.02 | 1.12 | -2.30 | 0.04 | 0.21 | -0.81 | ✗ | 2.28 | 1.78 | 1.18 | 2.48 | 2.94 | 0.66 | ✗ |
| Bi-Anchor(Grad) | 2.92 | -0.51 | -1.49 | 0.26 | 0.54 | -2.38 | ✗ | 2.92 | 2.90 | 0.70 | 2.26 | 2.70 | -0.22 | ✗ |
| Self-Inf-N | -1.03 | -0.51 | 2.28 | -1.24 | 1.01 | 2.04 | ✗ | 2.90 | 3.32 | 2.45 | 1.21 | 0.69 | -2.21 | ✗ |
| LARF | -5.65 | -3.66 | 1.41 | 0.90 | 3.23 | **8.88** | ✗ | 1.86 | 1.17 | 3.04 | 3.38 | 3.21 | 1.35 | ✗ |
| **SQSD(Aegis)** | -1.78 | 0.03 | 1.36 | 1.81 | 3.08 | 4.86 | ✓ | -1.21 | 1.22 | 2.22 | 2.45 | 3.70 | 4.91 | ✓ |
| **SQSD(Beaver)** | -4.00 | -0.62 | 1.35 | 2.53 | 3.70 | 7.70 | ✓ | -2.10 | 1.17 | 0.91 | 2.58 | 3.43 | **5.53** | ✗ |
| *Llama3.1-8B-Instruct* | | | | | | | | | | | | | | |
| Reward Model | -3.68 | -2.12 | 0.34 | -2.24 | -0.89 | 2.79 | ✗ | -3.46 | 1.17 | -0.32 | 1.23 | 1.86 | 5.32 | ✗ |
| Bi-Anchor(Reps) | -1.85 | -1.93 | -2.40 | -2.73 | -0.95 | 0.90 | ✗ | 0.45 | 0.94 | -0.56 | 0.18 | -1.50 | -1.95 | ✗ |
| Bi-Anchor(Grad) | -2.39 | -3.04 | -1.79 | -0.60 | 0.57 | 2.96 | ✗ | 0.13 | -0.03 | 0.52 | -1.22 | -0.47 | -0.60 | ✗ |
| Self-Inf-N | 0.12 | -1.37 | -3.02 | -3.73 | -1.17 | -1.29 | ✗ | -1.56 | 0.09 | -0.67 | -2.34 | -1.25 | 0.31 | ✗ |
| LARF | -4.29 | -4.28 | -1.10 | -1.64 | 0.99 | 5.28 | ✗ | -3.64 | -0.32 | -0.69 | 1.74 | 2.18 | 5.82 | ✗ |
| **SQSD(Aegis)** | -4.76 | 0.23 | -1.91 | 0.27 | 3.35 | **8.11** | ✗ | -2.23 | -0.09 | 0.18 | 0.40 | 1.49 | 3.72 | ✓ |
| **SQSD(Beaver)** | -4.93 | -3.64 | -2.11 | 1.63 | 1.73 | 6.66 | ✓ | -3.05 | -0.48 | 0.13 | 1.02 | 3.35 | **6.40** | ✓ |
| *Llama2-7b-Chat* | | | | | | | | | | | | | | |
| Reward Model | -1.21 | 2.83 | 3.37 | 4.39 | 5.66 | 6.87 | ✓ | 2.24 | 3.90 | 4.63 | 5.52 | 5.29 | 3.05 | ✗ |
| Bi-Anchor(Reps) | 2.50 | 2.31 | 3.69 | 5.14 | 2.92 | 0.42 | ✗ | 3.26 | 2.61 | 3.97 | 5.22 | 5.20 | 1.94 | ✗ |
| Bi-Anchor(Grad) | 1.98 | 3.42 | 3.29 | 3.96 | 2.64 | 0.66 | ✗ | 4.34 | 4.24 | 4.82 | 4.35 | 3.65 | -0.69 | ✗ |
| Self-Inf-N | 3.95 | 3.98 | 4.24 | 3.80 | 4.50 | 0.55 | ✗ | 3.83 | 4.96 | 4.30 | 4.24 | 4.01 | 0.18 | ✗ |
| LARF | 4.39 | 4.85 | 1.98 | 2.79 | 2.99 | -1.40 | ✗ | 4.36 | 4.18 | 4.24 | 3.71 | 4.25 | -0.11 | ✗ |
| **SQSD(Aegis)** | -1.25 | 2.00 | 3.17 | 3.78 | 5.48 | 6.73 | ✓ | -0.33 | 4.07 | 4.45 | 4.80 | 5.24 | **5.57** | ✓ |
| **SQSD(Beaver)** | -1.15 | 1.57 | 3.00 | 3.84 | 6.00 | **7.15** | ✓ | 0.47 | 4.75 | 4.21 | 5.12 | 5.39 | 4.92 | ✗ |

*Table 12.* Effectiveness of SQSD on AdvBench. ASR (%) on AdvBench for models fine-tuned on risk-ranked subsets by various methods. S1-S5 represent 1000 samples each, uniformly sampled from highest to lowest risk rankings. Δ denotes ASR difference (S1 - S5). Mono indicates whether ASR decreases monotonically across subsets (✓ is yes).

| Method | Dolly | | | | | | | Alpaca | | | | | | |
|---|---|---|---|---|---|---|---|---|---|---|---|---|---|---|
| | S1 | S2 | S3 | S4 | S5 | Δ ↑ | Mono | S1 | S2 | S3 | S4 | S5 | Δ ↑ | Mono |
| *Qwen3-8B* | | | | | | | | | | | | | | |
| Reward Model | 46.73 | 15.00 | 2.69 | 6.54 | 1.35 | 45.38 | ✗ | 77.12 | 6.92 | 13.27 | 12.31 | 9.81 | **67.31** | ✗ |
| Bi-Anchor(Reps) | 8.27 | 8.08 | 8.65 | 21.73 | 10.00 | -1.73 | ✗ | 11.35 | 29.04 | 43.08 | 16.92 | 6.92 | 4.43 | ✗ |
| Bi-Anchor(Grad) | 3.85 | 10.96 | 20.19 | 11.92 | 14.81 | -10.96 | ✗ | 5.58 | 11.73 | 28.65 | 9.42 | 20.77 | -15.19 | ✗ |
| Self-Inf-N | 3.65 | 0.96 | 13.08 | 14.62 | 9.04 | -5.39 | ✗ | 2.69 | 2.69 | 13.27 | 18.85 | 43.27 | -40.58 | ✗ |
| LARF | 50.19 | 22.88 | 5.96 | 0.77 | 0.19 | 50.00 | ✓ | 39.04 | 23.85 | 7.69 | 5.38 | 3.85 | 35.19 | ✓ |
| **SQSD(Aegis)** | 38.65 | 18.85 | 6.35 | 2.88 | 0.77 | 37.88 | ✓ | 57.12 | 24.62 | 14.62 | 7.12 | 2.69 | 54.43 | ✓ |
| **SQSD(Beaver)** | 73.65 | 18.65 | 12.12 | 4.81 | 0.58 | **73.07** | ✓ | 70.00 | 17.69 | 13.65 | 8.65 | 2.88 | 67.12 | ✗ |
| *Llama3.1-8B-Instruct* | | | | | | | | | | | | | | |
| Reward Model | 73.27 | 47.12 | 15.77 | 38.65 | 21.54 | 51.73 | ✗ | 75.96 | 2.12 | 22.12 | 4.04 | 11.92 | 64.04 | ✗ |
| Bi-Anchor(Reps) | 39.42 | 45.58 | 42.88 | 36.92 | 19.62 | 19.80 | ✗ | 7.50 | 5.19 | 14.62 | 10.96 | 54.04 | -46.54 | ✗ |
| Bi-Anchor(Grad) | 62.50 | 57.12 | 36.73 | 17.31 | 6.15 | 56.35 | ✓ | 11.92 | 16.15 | 7.88 | 35.00 | 31.54 | -19.62 | ✗ |
| Self-Inf-N | 2.88 | 44.42 | 47.31 | 54.23 | 15.96 | -13.08 | ✗ | 86.92 | 14.62 | 14.62 | 63.65 | 81.15 | 5.77 | ✗ |
| LARF | 56.92 | 79.23 | 17.31 | 35.38 | 2.69 | 54.23 | ✗ | 74.23 | 37.88 | 26.73 | 2.50 | 8.46 | 65.77 | ✗ |
| **SQSD(Aegis)** | 90.19 | 18.08 | 31.92 | 5.77 | 4.81 | 85.38 | ✗ | 68.46 | 21.15 | 11.92 | 8.27 | 5.58 | 62.88 | ✓ |
| **SQSD(Beaver)** | 89.23 | 58.65 | 33.65 | 4.62 | 1.15 | **88.08** | ✓ | 77.50 | 10.19 | 9.04 | 3.27 | 0.38 | **77.12** | ✓ |
| *Llama2-7b-Chat* | | | | | | | | | | | | | | |
| Reward Model | 58.27 | 1.73 | 1.73 | 0.19 | 0.19 | **58.08** | ✓ | 35.00 | 2.69 | 1.73 | 0.58 | 0.58 | 34.42 | ✓ |
| Bi-Anchor(Reps) | 5.38 | 7.50 | 0.77 | 0.58 | 8.08 | -2.70 | ✗ | 11.54 | 9.62 | 3.85 | 4.81 | 13.08 | -1.54 | ✗ |
| Bi-Anchor(Grad) | 15.19 | 0.96 | 0.77 | 0.77 | 2.69 | 12.50 | ✗ | 14.04 | 7.12 | 2.12 | 3.46 | 6.54 | 7.50 | ✗ |
| Self-Inf-N | 3.85 | 0.96 | 0.58 | 4.23 | 0.38 | 3.47 | ✗ | 7.69 | 0.58 | 5.96 | 12.88 | 8.08 | -0.39 | ✗ |
| LARF | 1.35 | 0.58 | 6.54 | 1.15 | 0.58 | 0.77 | ✗ | 7.31 | 1.73 | 3.46 | 10.96 | 3.65 | 3.66 | ✗ |
| **SQSD(Aegis)** | 37.31 | 4.23 | 1.54 | 0.96 | 0.00 | 37.31 | ✓ | 53.27 | 3.85 | 2.50 | 1.15 | 0.96 | **52.31** | ✓ |
| **SQSD(Beaver)** | 35.00 | 7.31 | 2.31 | 0.77 | 0.00 | 35.00 | ✓ | 29.81 | 5.96 | 3.65 | 0.38 | 1.15 | 28.66 | ✗ |

*Table 13.* Effectiveness of SQSD on HEx-PHI. ASR (%) on HEx-PHI for models fine-tuned on risk-ranked subsets by various methods. S1-S5 represent 1000 samples each, uniformly sampled from highest to lowest risk rankings. $\Delta$ denotes ASR difference between (S1 - S5). Mono indicates whether ASR decreases monotonically across subsets (✓ is yes).

| Method | Dolly | | | | | | | Alpaca | | | | | | |
|---|---|---|---|---|---|---|---|---|---|---|---|---|---|---|
| | S1 | S2 | S3 | S4 | S5 | $\Delta\uparrow$ | Mono | S1 | S2 | S3 | S4 | S5 | $\Delta\uparrow$ | Mono |
| *Qwen3-8B* | | | | | | | | | | | | | | |
| Reward Model | 63.45 | 30.34 | 16.55 | 15.17 | 9.31 | 54.14 | ✓ | 66.55 | 17.59 | 30.34 | 21.72 | 28.97 | 37.58 | ✗ |
| Bi-Anchor(Reps) | 31.38 | 22.41 | 26.21 | 40.34 | 27.59 | 3.79 | ✗ | 28.28 | 37.24 | 37.93 | 27.59 | 17.93 | 10.35 | ✗ |
| Bi-Anchor(Grad) | 19.66 | 32.07 | 33.45 | 30.69 | 21.38 | -1.72 | ✗ | 17.93 | 20.00 | 33.45 | 23.45 | 33.45 | -15.52 | ✗ |
| Self-Inf-N | 19.66 | 12.76 | 22.41 | 28.97 | 26.21 | -6.55 | ✗ | 13.10 | 13.79 | 30.34 | 35.86 | 52.41 | -39.31 | ✗ |
| LARF | 54.14 | 44.48 | 18.97 | 10.00 | 5.52 | 48.62 | ✓ | 40.00 | 35.52 | 19.31 | 19.66 | 16.90 | 23.10 | ✗ |
| **SQSD(Aegis)** | 68.97 | 47.24 | 27.24 | 14.14 | 12.07 | 56.90 | ✓ | 57.24 | 35.52 | 33.10 | 22.07 | 14.83 | 42.41 | ✓ |
| **SQSD(Beaver)** | 82.76 | 41.38 | 27.59 | 17.93 | 11.03 | **71.73** | ✓ | 61.38 | 28.97 | 27.59 | 18.28 | 11.03 | **50.35** | ✓ |
| *Llama3.1-8B-Instruct* | | | | | | | | | | | | | | |
| Reward Model | 81.38 | 47.59 | 25.17 | 44.14 | 24.48 | 56.90 | ✗ | 79.66 | 15.52 | 35.17 | 17.59 | 36.21 | 43.45 | ✗ |
| Bi-Anchor(Reps) | 48.62 | 48.62 | 56.90 | 52.76 | 16.90 | 31.72 | ✗ | 18.62 | 17.93 | 30.69 | 26.55 | 59.66 | -41.04 | ✗ |
| Bi-Anchor(Grad) | 71.03 | 66.55 | 47.24 | 22.76 | 16.55 | 54.48 | ✓ | 37.24 | 38.97 | 24.83 | 51.72 | 35.52 | 1.72 | ✗ |
| Self-Inf-N | 15.86 | 57.93 | 69.31 | 66.21 | 19.66 | -3.80 | ✗ | 86.21 | 36.90 | 30.34 | 65.86 | 60.34 | 25.87 | ✗ |
| LARF | 70.00 | 79.31 | 31.38 | 38.28 | 8.28 | 61.72 | ✗ | 71.38 | 59.31 | 59.31 | 8.62 | 20.34 | 51.04 | ✗ |
| **SQSD(Aegis)** | 82.76 | 26.55 | 52.41 | 25.86 | 10.69 | 72.07 | ✗ | 81.38 | 34.48 | 23.10 | 26.90 | 12.76 | 68.62 | ✓ |
| **SQSD(Beaver)** | 90.00 | 61.38 | 42.07 | 8.62 | 2.76 | **87.24** | ✓ | 82.41 | 27.93 | 26.55 | 10.69 | 1.38 | **81.03** | ✓ |
| *Llama2-7b-Chat* | | | | | | | | | | | | | | |
| Reward Model | 54.14 | 12.76 | 12.76 | 1.72 | 1.38 | 52.76 | ✓ | 50.34 | 16.90 | 16.90 | 7.59 | 9.31 | 41.03 | ✗ |
| Bi-Anchor(Reps) | 25.86 | 31.03 | 18.97 | 4.83 | 26.21 | -0.35 | ✗ | 26.55 | 36.21 | 21.38 | 25.86 | 34.14 | -7.59 | ✗ |
| Bi-Anchor(Grad) | 33.45 | 11.72 | 13.10 | 7.24 | 22.76 | 10.69 | ✗ | 29.66 | 24.14 | 18.97 | 16.90 | 32.07 | -2.41 | ✗ |
| Self-Inf-N | 21.03 | 13.45 | 10.69 | 20.69 | 10.69 | 10.34 | ✗ | 16.90 | 13.10 | 24.48 | 37.59 | 25.86 | -8.96 | ✗ |
| LARF | 8.97 | 16.90 | 31.38 | 14.48 | 4.83 | 4.14 | ✗ | 19.31 | 17.24 | 19.31 | 32.41 | 25.17 | -5.86 | ✗ |
| **SQSD(Aegis)** | 58.97 | 35.17 | 20.00 | 8.28 | 4.83 | 54.14 | ✓ | 57.59 | 27.24 | 19.31 | 11.03 | 10.00 | **47.59** | ✓ |
| **SQSD(Beaver)** | 53.79 | 30.69 | 20.69 | 6.21 | 4.48 | **49.31** | ✓ | 47.59 | 24.83 | 19.66 | 7.24 | 8.62 | 38.97 | ✗ |

