# OpenReview forum: "From Parameter Dynamics to Risk Scoring: Quantifying Sample-Level Safety Degradation in LLM Fine-tuning"
_ICML.cc/2026/Conference — ICML 2026 regular_

### Official Review · Reviewer_PGR4 · 2026-03-08

**Soundness:** 3
**Presentation:** 3
**Significance:** 3
**Originality:** 2
**Overall Recommendation:** 4
**Confidence:** 4

**Summary:**

This paper investigates the phenomenon of safety alignment degradation in Large Language Models (LLMs) during fine-tuning on benign data. A central concept analyzed by this study is the dynamic trajectory of model parameters during fine-tuning, specifically how they cumulatively drift toward danger-aligned directions. By tracking parameter dynamics, the authors reveal the correlation between safety degradation and parameter drift toward dangerous directions, and based on this, propose the Sample-Level Quantification of Safety Degradation (SQSD) method. This method assigns continuous risk scores to each training sample by calculating the projection difference of sample-induced parameter updates on danger versus safety directions. Overall, the submission's main aspect is providing a mechanistic explanation for safety degradation and a practical tool for risk-aware data selection, demonstrating strong transferability across model architectures and scales.

**Compliance With Llm Reviewing Policy:**

Affirmed.

**Final Justification:**

The rebuttal satisfactorily resolved my main concerns, and I am happy to upgrade my recommendation to Accept.

**Key Questions For Authors:**

1.  The core formula of the paper (gradient inner product approximation) is mathematically highly similar to Influence Function approximations under the assumption that the Hessian matrix is an identity matrix. Could the authors add a discussion in the Related Work or Method section to explicitly compare the similarities and differences between SQSD and classical IF approximations (e.g., $\nabla_\theta L(z)^\top \nabla_\theta L(z_{test})$)? In particular, what are the specific reasons why raw IF approximations fail in safety degradation scenarios (e.g., the length bias issue shown in Appendix H)?


2.  The paper currently mainly explains the mechanism of parameter drift but does not deeply explain the root cause of why benign samples induce such drift. Have the authors explored the objective conflict between "Instruction Following (Compliance)" and "Safety Refusal"? Have you analyzed semantic commonalities in high-risk samples (e.g., are specific topics more likely to trigger dangerous directions)?

3.  The paper lacks a comparison of computational costs with baseline methods. Could the authors supplement the wall-clock time and GPU memory usage required to compute SQSD scores for 1k samples and compare this with the inference time of Reward Models? Additionally, what is the specific cost of "Pilot Fine-tuning" to find sensitive checkpoints?

**Limitations:**

yes

**Strengths And Weaknesses:**

1.Soundness

   Strengths: The experimental design is rigorous and sufficiently validated. The authors not only verify the consistency of the parameter drift mechanism across multiple models (Qwen, LLaMA) and datasets (Dolly, Alpaca) but also demonstrate the necessity of SQSD components through detailed ablation studies (e.g., module-wise normalization, bidirectional contrastive design). The Taylor approximation theory provides a mathematical connection between parameter updates and output preferences, enhancing the interpretability of the method.

   Weaknesses: The core formula (gradient inner product approximation) is mathematically highly similar to Influence Functions (IF) approximations under the assumption that the Hessian matrix is an identity matrix. The paper does not sufficiently discuss this theoretical boundary, which may raise questions about methodological novelty. Although the authors address specific biases through normalization and bidirectional contrast, they need to argue more clearly why raw IF approximations fail in this scenario (e.g., the length bias issue shown in Appendix H).

 2.Presentation

   Strengths: The paper structure is clear and the logic flows smoothly. The narrative is coherent, moving from phenomenon observation (parameter drift) to method proposal (SQSD), and then to experimental validation. Figures (e.g., Figure 2 parameter trajectory plot) intuitively display the core findings.

   Weaknesses: The discussion on the root cause of safety degradation is somewhat superficial. The paper mainly stays at the level of phenomenon description (Mechanism) regarding "parameter drift toward dangerous directions," without deeply explaining the semantic-level root causes (Root Cause) of why benign samples induce such drift.

 3.Significance

   Strengths: It addresses a key pain point in the LLM safety community: safety degradation caused by benign fine-tuning is difficult to prevent. The continuous risk scores provided by SQSD are more practical than existing discrete subset selection methods, supporting finer-grained data cleaning. Validation of transferability across models and scales indicates broad deployment potential for this method.

   Weaknesses: There is a lack of computational cost and efficiency analysis. The paper claims SQSD does not require fine-tuning n times, but it still requires computing gradients for each sample and finding sensitive initialization points. Currently, there is no comparison of computational costs with baseline methods (e.g., Reward Model), making it difficult to assess practical deployment feasibility.

4. Originality

 Strengths: Revealing the safety degradation mechanism from a parameter dynamics perspective is a novel entry point, distinct from existing static comparison studies. Proposing a continuous risk quantification paradigm rather than discrete selection avoids the "boundary collapse" problem, representing a methodological innovation.

   Weaknesses: As mentioned above, the core computation logic of SQSD overlaps with gradient similarity/Influence Function methods. If the distinction from existing Influence Function variants cannot be clearly defined, the originality will be compromised. Additionally, the analysis of semantic conflicts regarding "why benign samples align with dangerous directions" is insufficient, limiting breakthroughs in theoretical depth.

---

> ### Author Rebuttal · Authors · 2026-03-28
>
> ## Q1: Relationship between SQSD and Influence Functions
>
> ### 1). Mathematical Equivalence of IF, Bi-Anchor, and Learning Dynamics
> Our theoretical foundation extends Learning Dynamics [1] and Bi-Anchor(Grad), which share the same mathematical form as IF approximations under $H^{-1} = I$:
>
> $$\ell(z_{test};\theta) - \ell(z_{test};\theta') \approx k \nabla_\theta \ell(z_{train};\theta)^\top \nabla_\theta \ell(z_{test};\theta) \tag{1}$$
>
> But their derivations differ: Classical IF derives the parameter change after removing sample $z$ via the implicit function theorem: $\theta' = \theta_{-z} \approx \theta - \frac{1}{n} H_{\theta}^{-1} \nabla_\theta \mathcal{L}(z, \theta)$, and reaches Eq. (1) need $H^{-1} \approx I$; Bi-Anchor (Grad) reaches it via a single gradient step $\theta' = \theta - \eta \nabla_\theta \mathcal{L}(z, \theta)$. The final form is identical but the assumptions differ.
>
> [1] Learning Dynamics of LLM Finetuning (ICLR2025)
>
> ### 2). Key Distinction: SQSD vs. IF
> Both IF and Bi-Anchor (Grad) use Eq. (1) to approximate the loss change on $z_{test}$ caused by the parameter update induced by $z_{train}$. SQSD instead fits the difference in loss of $z$ evaluated at $\theta_{ref}$ versus $\theta_{target}$ via the similarity between the sample-induced parameter update and the parameter displacement:
>
> $$\eta \left[ \ell(z, \theta_{ref}) - \ell(z, \theta_{target}) \right] \approx (\theta' - \theta_{ref})^\top (\theta_{target} - \theta_{ref}) \tag{2}$$
>
> By instantiating $\theta_{target}$ as $\theta_{danger}$ or $\theta_{safety}$, Eq. (2) approximates the degree to which $z_{train}$ prefers the dangerous or safe output distribution over the reference one.
>
> ### 3). Why IF Approximations Fail for Safety Risk Quantification
> Response-length bias affects all gradient-based methods with unnormalized updates, not specific to IF. The more fundamental failures are:
>
> **Reason1: SFT loss on anchor samples does not reflect safety behavior.** Fine-tuning on 1k BeaverTails(safe) samples increases ASR from 8.18% to 46.91% (Qwen3-8B), showing that SFT loss on anchor samples reflects surface-level content fit rather than safety behavior.
> **Reason2: Sparse anchor coverage.** Bi-Anchor(Grad), the IF-equivalent baseline, uses only 10 anchor samples, insufficient to cover the multi-dimensional safety space. SQSD's directions aggregate 3k dangerous samples and 10k preference pairs, ensuring broader coverage.
>
>
> ## Q2: Root Cause Analysis of Safety Degradation
>
> **1）The objective conflict hypothesis.** We clarify that the safety degradation studied in this paper is a **continual learning forgetting problem** rather than a **multi-objective conflict problem**. We measure the instruction-following (IF) ability of S1-S5 fine-tuned models (Qwen3, Beaver):
>
> ||Base|S1|S2|S3|S4|S5|
> |-|-|-|-|-|-|-|
> |IF (Alpaca)|0.82|0.64|0.64|0.65|0.69|0.72|
> |ASR% (Alpaca)|8.18|50.91|19.09|18.73|7.27|3.27|
> |IF (Dolly)|0.82|0.52|0.56|0.58|0.67|0.75|
> |ASR% (Dolly)|8.18|71.27|29.45|10.18|7.27|2.55|
>
> Analysis: (1) SFT on a well-aligned model readily induces forgetting, even instruction-tuning data degrades both safety and instruction-following, with high-risk samples accelerating this; (2) the two forms of forgetting are **synchronous**, both improve monotonically from S1 to S5, directly contradicting the objective conflict hypothesis. **Notably, this co-forgetting pattern suggests that SQSD may generalize beyond safety to quantifying forgetting across different capability domains.**
>
> **2）Semantic commonalities of high-risk samples.** We analyze the category distribution on Dolly(Llama3, Beaver):
>
> |Category|S1|S2|S3|S4|S5|
> |-|-|-|-|-|-|
> |creative_writing|172|49|21|15|16|
> |brainstorming|291|165|97|70|45|
> |general_qa|268|182|127|84|76|
> |classification|9|62|98|217|414|
>
> Open-ended tasks(`creative_writing`, `brainstorming`, `general_qa`) concentrate in S1 while structured tasks (`classification`) concentrate in S5 (consistent across all configurations), suggesting that open-ended tasks with ill-defined answer boundaries are more prone to inducing safety degradation.
>
> **3）Why benign samples induce danger-direction drift.** As shown by experiments above and Reviewer Akkt Q1, the danger direction primarily encodes alignment forgetting. When train samples exhibit large distributional gaps from the aligned model's output, fitting these gaps will erodes the model's alignment, manifesting as danger-direction drift.
>
> ## Q3: Computational Cost
> Overhead for scoring 1k samples:
>
> ||Reward Model|Bi-Anchor(Reps)|Self-Inf-N|Bi-Anchor(Grad)|LARF|Ours|
> |-|-|-|-|-|-|-|
> |Memory(G)|25|32|45|90|23|44|
> |Time(s)|52|391|2718|8127|72|1918|
>
> Ours is the most efficient gradient-based method, 1918s covers two gradient passes (danger and safety), merging into one pass can halve this cost with larger GPU memory.
> Pilot fine-tuning is a one-time cost of ~7h (2h training, 5h evaluation), reusing checkpoints from §5.2, evaluating only later-stage checkpoints can further reduce this.

---

> > ### Author Rebuttal · Reviewer_PGR4 · 2026-04-02
> >
> > I appreciate the authors’ detailed and data-backed rebuttal. I have no further questions. Specifically:
> >
> > The distinction between SQSD and Influence Functions (Q1) has been clarified, and the empirical evidence regarding “SFT loss ≠ safety behavior” is convincing.
> >
> > The “co-forgetting” hypothesis and semantic analysis (Q2) provide valuable insights into the underlying mechanisms, effectively addressing my concern regarding superficial analysis.
> >
> > The computational cost comparison (Q3) demonstrates the method’s practical feasibility relative to other gradient-based approaches.

---

> > > ### Author Response · Authors · 2026-04-02
> > >
> > > We sincerely thank the reviewer for the constructive feedback throughout the discussion and for confirming that all concerns have been fully resolved. Your questions and suggestions have been highly valuable in improving both the clarity and depth of our work. We will carefully incorporate the additional analyses and clarifications from the rebuttal into the final manuscript.
> > >
> > > If you feel that the revisions and additional evidence have addressed your concerns to a satisfactory level, we would be grateful if you could kindly consider reflecting this in your final assessment. We fully respect your judgment and appreciate your time and effort in reviewing our work.
> > >
> > > Should anything remain unclear or if further questions arise, we would be very happy to continue the discussion.

---

### Official Review · Reviewer_btH8 · 2026-03-08

**Soundness:** 3
**Presentation:** 3
**Significance:** 3
**Originality:** 3
**Overall Recommendation:** 4
**Confidence:** 3

**Summary:**

This paper studies why safety alignment in large language models can be easily weakened during benign fine-tuning. Rather than only comparing model parameters or hidden states before and after training, the authors analyze the evolution of parameters during the fine-tuning process. Their analysis suggests that benign samples can gradually push parameters toward directions associated with unsafe behavior. Based on this observation, the paper proposes Sample-Level Quantification of Safety Degradation (SQSD), which estimates the safety risk of individual training samples by measuring how strongly their parameter updates align with safety or danger directions. The method assigns a continuous risk score to each sample and is supported by a theoretical interpretation using a first-order Taylor approximation. Experiments on multiple models and datasets indicate that SQSD can effectively identify high-risk samples and generalize across architectures, model sizes, and PEFT settings.

**Compliance With Llm Reviewing Policy:**

Affirmed.

**Key Questions For Authors:**

-  The experiments mainly evaluate SQSD under fine-tuning on general instruction datasets such as Alpaca and Dolly. It would be interesting to see whether the same behavior holds for domain-specific datasets (e.g., medical or legal data), where benign fine-tuning might exhibit different safety dynamics. Results on such datasets would strengthen the robustness claims.

- SQSD requires computing a one-step gradient update for each training sample. For large datasets and models this could be expensive. Some discussion or analysis of the computational cost (runtime or memory overhead) would be helpful.

- Recent work such as Qi et al. (2025) proposes strong defenses against safety degradation during fine-tuning by modifying the training loss. Although that work focuses on full-parameter fine-tuning, its core idea should also be compatible with parameter-efficient methods such as LoRA. Could the authors provide a comparison between SQSD and Qi et al. (2025) in terms of performance on safety degradation?

References:

1.Qi et al., "Safety Alignment Should be Made More Than Just a Few Tokens Deep", ICLR 2025

**Limitations:**

yes

**Strengths And Weaknesses:**

**Strengths**

-  The paper is clearly written and easy to follow. The motivation and the proposed method are presented in a fairly straightforward way.

- Experiments cover several model architectures and parameter scales, which gives some confidence that the method is not tied to a single model or setup.



**Weaknesses**

- SQSD requires computing the parameter update induced by each training sample through one-step gradient descent. For large datasets or large models, this per-sample gradient computation could be quite expensive in practice. Some discussion of the computational cost would be helpful.

-  The method depends on estimating safety and danger directions in parameter space. It is not clear how sensitive the results are to the choice of reference models or alignment settings used to obtain these directions. Additional analysis of the stability of these directions across different setups would strengthen the paper.

---

> ### Author Rebuttal · Authors · 2026-03-30
>
> We thank the reviewer for the detailed and constructive comments. These suggestions have helped us strengthen the paper. We address each point below.
> ## Q1: Sensitivity to Reference Models and Alignment Settings
>
> **1) The reference model has no degree of freedom.** Safety and danger directions are parameter-space displacement vectors (Eqs. 3–4), whose dimensionality must exactly match the target model's architecture. Therefore, the reference model used for direction construction is necessarily the target model itself — there is no choice of "which reference model" involved in our pipeline.
>
> **2) The core challenge is initialization sensitivity, not direction fragility.** SQSD computes a sample's risk score via the projection of its one-step gradient update onto safety-relevant directions. Since this gradient is evaluated at a specific parameter state, the resulting projection reflects the local geometry at that state. In high-dimensional parameter space, a given direction cannot maintain uniform sensitivity across all parameter states, the same direction that strongly separates sample risks at one state may yield weak discrimination at another (Figure 5). Therefore, the key to effective SQSD is not finding a universally robust direction, but selecting an initialization state where the model's parameters are locally sensitive to the given direction (§4.3, Appendix E). We propose two initialization paths: drift-enhanced sensitivity, which leverages fine-tuning checkpoints where cumulative drift has placed parameters in a sensitive region, and linear-path sensitivity, which interpolates along the direction to locate a sensitive point. Taking Llama-3.1-8B-Instruct (Dolly, Beaver) as an example, we compare five initialization states spanning both strategies:
>
> |Init Strategy|Init State|S1|S2|S3|S4|S5|Mono|
> |-|-|-|-|-|-|-|-|
> |Drift-enhanced|Ckpt 5700 (Top1)|79.82%|57.27%|37.45%|5.27%|4.73%|✓|
> ||Ckpt 4800 (Top2)|79.45%|61.27%|41.09%|16.18%|5.45%|✓|
> ||Ckpt 4350 (Top3)|77.45%|65.82%|40.18%|18.73%|8.00%|✓|
> ||Ckpt 4650 (Top4)|74.00%|53.82%|32.91%|20.36%|2.55%|✓|
> |Linear-path|α=1|75.64%|39.64%|35.82%|27.82%|8.73%|✓|
>
> All five settings maintain monotonically decreasing ASR, with S1 consistently at 74–80% and S5 at 2–9%, demonstrating that SQSD produces reliable risk predictions across different initialization strategies and states. Higher-sensitivity initializations further improve the discriminability of intermediate-risk samples (e.g., Top1 Δ=75.09% vs Linear-path Δ=66.91%).
>
> **3) Direction construction is empirically robust.** Two danger directions from entirely different data sources (Aegis-unsafe from Ghosh et al., 2024 and Beaver-unsafe from Ji et al., 2023) both achieve consistent monotonic ASR decrease across nearly all configurations (Table 1: 10/12 settings). This indicates that SQSD is not sensitive to the specific alignment data used for direction construction.
>
> ## Q2:Generalization to Domain-Specific Datasets
>
> We evaluate SQSD on three datasets with distinct characteristics (Qwen3-8B, Beaver):
>
> |Dataset|S1|S2|S3|S4|S5|Mono|
> |-|-|-|-|-|-|-|
> |Role-Play|40.73|34.55|14.73|6.73|1.64|✓|
> |BrainStorming|4.18|2.73|2.00|1.82|0.91|✓|
> |Medical|0.10|0.11|0.10|0.20|0.11|✗|
>
> SQSD maintains monotonically decreasing ASR under both strong (Role-Play) and weak (BrainStorming) safety degradation. Medical ASR stays at 0.1–0.2%, indicating the data itself does not induce meaningful safety degradation, risk quantification is beyond the reach of any sample-level method.
>
> # Q3: Computational Cost
>
> Cost for scoring 1k samples:
>
> | | Reward Model | Bi-Anchor(Reps) | Self-Inf-N | Bi-Anchor(Grad) | LARF | Ours |
> |-|-|-|-|-|-|-|
> | Memory(G) | 25 | 32 | 45 | 90 | 23 | 44 |
> | Time(s) | 52 | 391 | 2718 | 8127 | 72 | 1918 |
>
> SQSD is the most efficient gradient-based method. This 1918s covers two gradient passes (danger and safety), a single merged pass can halve this cost.
>
> ## Q4:Comparison with Qi et al. (2025)
>
> Qi et al. modifies the training loss to achieve deeper safety alignment, operating as a training-stage defense; SQSD is a data-level risk quantification method. The two serve different purposes but are complementary. We compare them under the same setup (Qwen3-8B, Dolly, 3000 samples):
>
> | |Standard SFT|Qi et al. (2025)|SQSD (filter)|
> |-|-|-|-|
> |Data|3000|3000|2500|
> |ASR(%)|30.18|4.18|4.00|
>
> SQSD achieves comparable performance to Qi et al. (4.00% vs 4.18%) by simply removing 500 high-risk samples, without modifying the training procedure. The two can be combined: SQSD's continuous risk scores can guide per-sample safety constraints, applying stronger regularization to high-risk samples and lighter constraints to low-risk ones to preserve task performance.

---

> > ### Author Rebuttal · Reviewer_btH8 · 2026-04-02
> >
> > Thank you for your response. Your clarifications have resolved my previous concerns. The paper is generally in good shape. I'm happy to accept this paper.

---

> > > ### Author Response · Authors · 2026-04-05
> > >
> > > Thank you for your positive feedback and for confirming that your concerns have been resolved. We are glad the clarifications were helpful, and the additional analyses will be incorporated into the revised manuscript.
> > >
> > > Thank you again for the thoughtful review.

---

### Official Review · Reviewer_zzD9 · 2026-03-08

**Soundness:** 3
**Presentation:** 2
**Significance:** 3
**Originality:** 3
**Overall Recommendation:** 4
**Confidence:** 3

**Summary:**

The paper studies the harmful fine-tuning attack on LLMs. The paper analyze the dynamics of parameter updates during finetuning. The main observation is model parameters gradually drift toward unsafe directions in parameter space while remaining largely orthogonal to safety-aligned directions. Based on the observation, the paper proposes SQSD method.

**Compliance With Llm Reviewing Policy:**

Affirmed.

**Final Justification:**

No more questions

**Key Questions For Authors:**

See above

**Limitations:**

See above

**Strengths And Weaknesses:**

The paper covers mechanistic perspective on safety degradation by analyzing training dynamics. I like that.
SQSD offers a practical way to estimate which samples are most likely to degrade safety.

While parameter drift correlates with safety degradation, I'm curious if the drift causes the degradation rather than accompanying it. In other words, is this correlation or causal proof?

What is the computational cost for gradient based estimation?

Can you compare your method with some of the latest defense methods?
1. Booster: Tackling harmful fine-tuning for large language models via attenuating harmful perturbation
2. Shape it up! restoring llm safety during finetuning
3. Rebellion: Noise-Robust Reasoning Training for Audio Reasoning Models

---

> ### Author Rebuttal · Authors · 2026-03-29
>
> # Q1: Correlation vs. Causation
>
> We thank the reviewer for this critical question. Our evidence goes beyond mere correlation, spanning multiple levels of causal reasoning. We also provide two new experiments to further strengthen the causal argument.
>
> **Existing causal evidence.** (1) Parameter steering experiments (§3.1, Appendix A.2) constitute direct causal interventions: perturbing parameters along the danger direction produces monotonic safety degradation across all configurations (Figure 5), directly demonstrating that danger-direction displacement *produces* safety degradation. (2) Figure 4 exhibits a dose-response relationship: larger data scale → greater cumulative danger-direction drift → more severe degradation (Safety Score: 3.8→-0.3). (3) SQSD scores are derived from the causal hypothesis that danger-direction drift drives degradation. The predictions are validated in 10/12 configurations (Table 1, ASR monotonically decreasing S1→S5), providing indirect causal support.
>
> **New Experiment 1: Causal mechanism verification.** Evidence (3) validates SQSD's risk ranking but does not verify whether these samples actually push parameters toward the danger direction. To test this, we fine-tune models on each subset (S1–S5) separately and compute the projection of parameter drift onto danger and safety directions. Taking Llama3 (Dolly, Aegis-Safe) as an example:
>
> ||S1|S2|S3|S4|S5|
> |-|-|-|-|-|-|
> |danger_aegis|1.694|0.827|0.609|0.281|0.054|
> |danger_beaver|2.179|1.634 |1.423|1.264|1.013|
> |safety_proj|-0.080|0.242|0.347|0.441|0.713|
> |ASR(%)|76.36|21.45|36.36|16.18|13.82|
>
> Danger projections decrease strictly monotonically from S1→S5, while safety projections increase. **This pattern is consistent across all 12 configurations**, completing the causal chain: high SQSD risk → larger danger-direction drift → more severe safety degradation.
>
> **New Experiment 2: Counterfactual intervention.** To test whether removing danger-direction drift can restore safety, we take a checkpoint that has already undergone safety degradation through benign fine-tuning, and project out the danger-direction component from its cumulative parameter displacement: $\Delta W^{\text{proj}} = \Delta W - \langle \Delta W, \hat{V}_{\text{danger}} \rangle \hat{V}_{\text{danger}}$.
>
> ||ASR (%)|
> |-|-|
> |Before|15.09|
> |After project out|0.91|
>
> ASR drops from 15.09% to 0.91%, directly demonstrating that safety degradation is encoded in the danger-direction component and removing it restores safety, proving causation rather than mere correlation.
>
> In summary, our evidence spans causal intervention, dose-response, predictive validation, mechanism verification, and counterfactual reasoning — collectively establishing that danger-direction drift causes safety degradation. We will include the new experiments and explicitly organize this causal argumentation in the revised manuscript.
>
> # Q2: Computational Cost
>
> Cost for scoring 1k samples:
>
> | | Reward Model | Bi-Anchor(Reps) | Self-Inf-N | Bi-Anchor(Grad) | LARF | Ours |
> |-|-|-|-|-|-|-|
> | Memory(G) | 25 | 32 | 45 | 90 | 23 | 44 |
> | Time(s) | 52 | 391 | 2718 | 8127 | 72 | 1918 |
>
> SQSD is the most efficient gradient-based method. The 1918s covers two gradient passes (danger and safety), a single merged pass can halve this cost.
>
> # Q3: Comparison with Defense Methods
>
> We thank the reviewer for suggesting these excellent works for comparison. These methods differ in objectives and threat models: SQSD quantifies fine-tuning risks in benign samples; Booster and STAR-DSS defend against harmful fine-tuning at the alignment and fine-tuning stages respectively; Rebellion targets audio model robustness. We apply SQSD for data filtering and compare with Booster and STAR-DSS. We apologize that Rebellion operates on a different modality and cannot be directly compared.
>
> **Setup** (Qwen3-8B, Dolly, 3000 random samples): Both reproduced from original repositories. SQSD removes 500 highest-risk samples, SFT on remaining 2500; Booster applies its alignment then fine-tunes on all 3000; STAR-DSS computes token (chunk)-level safety scores via LlamaGuard3 then weighted fine-tunes on all 3000.
>
> | Method | Data | ASR (%) |
> |-|-|-|
> | Standard SFT | 3000 | 30.18 |
> | STAR-DSS | 3000 | 29.12 |
> | Booster | 3000 | 6.36 |
> | SQSD (filter) | 2500 | 4.00 |
>
> **Analysis.** (1) SQSD achieves the lowest ASR by simply removing high-risk samples, while STAR-DSS shows negligible improvement. (2) STAR-DSS fails because LlamaGuard3 classifies 98% of tokens in benign data as safe (confidence > 0.9), degrading to standard SFT. SQSD captures risks invisible to safety classifiers and could be extended to token-level scoring for integration with STAR-DSS. (3) For Booster, we apologize for not fully aligning model settings with the original paper. Since Qwen3-8B is already well-aligned, additional alignment may have limited Booster's effectiveness.

---

> > ### Author Rebuttal · Reviewer_zzD9 · 2026-04-03
> >
> > Thanks! I'll keep my score

---

> > > ### Author Response · Authors · 2026-04-05
> > >
> > > We sincerely thank you for your positive assessment and for confirming that all concerns have been fully resolved. Your suggestions about “correlation or causal” helped us improve the paper, and we will include the corresponding additions in the revised manuscript. Thank you again for your thoughtful engagement throughout the discussion.

---

### Official Review · Reviewer_Akkt · 2026-03-09

**Soundness:** 3
**Presentation:** 3
**Significance:** 3
**Originality:** 3
**Overall Recommendation:** 5
**Confidence:** 3

**Summary:**

In this manuscript, the authors first analyze the cause of safety alignment degradation during benign data fine-tuning from the perspective of Parameter Dynamics. They reveal that this degradation is driven by the cumulative drift of model parameters toward the dangerous alignment direction during the training process. Based on this key insight, the authors propose a method called Sample-level Quantification of Safety Degradation (SQSD), which quantifies the fine-tuning risk of each training sample by measuring the difference between the projections of sample-induced parameter updates onto the dangerous and safe directions. The authors further validate the effectiveness of SQSD through extensive experiments, demonstrating its strong performance and transferability across different model architectures, parameter scales, and fine-tuning settings. The topic is significant for advancing LLM training and safety alignment. However, several aspects require further clarification and improvement to enhance the overall rigor and completeness of the paper.

**Compliance With Llm Reviewing Policy:**

Affirmed.

**Final Justification:**

The authors addressed my concerns about generalizability by conducting experiments on different types of data (explicitly harmful, safe, and high-risk benign) and performing SQSD ranking across 11 safety subcategories, demonstrating that the danger direction robustly captures various types of safety alignment forgetting.

The authors also report the computational cost.

At the same time, I also note that the authors addressed concerns raised by other reviewers through their experiments.

Accordingly, I raise my score to an accept.

**Key Questions For Authors:**

- The core idea of the method is to construct V_{danger} and V_{safety}. However, if the potentially risky patterns contained in the “benign” fine-tuning data (e.g., subtle biases) differ significantly from the explicit harmful data (e.g., violence or illegal advice) used to construct V_{danger}, it is unclear whether SQSD would still remain effective. In other words, is this “danger direction” in the parameter space sufficiently generalizable to cover different types of safety degradation?
- It is recommended to include a pseudocode description of the proposed method.
- It is recommended that the introduction of the baselines in Section 5.3 be moved to Section 5.1 (Experimental Setups), where it would be more appropriate.
- The experiments lack an analysis of computational overhead, such as runtime and peak GPU memory consumption for different methods.
- Analyze the limitations of this work.

**Strengths And Weaknesses:**

**Strengths**
- The manuscript starts from the issue of safety degradation that occurs when fine-tuning on benign data, analyzes the underlying causes, and proposes a sample-level parameter adjustment strategy to address this problem, which is an interesting and meaningful attempt.
- The experimental results show that the scores computed by SQSD can generalize across different model architectures, parameter scales, and fine-tuning methods, which is very important.

**Weaknesses**
- Please refer to the Key Questions for the Authors listed above.

---

> ### Author Rebuttal · Authors · 2026-03-30
>
> We thank the reviewer for the detailed and constructive comments, these suggestions have helped us strengthen the paper. We address each point below.
> ## Q1：Is the Danger Direction Sufficiently Generalizable to Cover Different Types of Safety Degradation?
>
> The question of whether the danger direction can cover all potential risk patterns in benign fine-tuning data has significant implications for extending SQSD to broader data scenarios.
>
>
> **(1)Safety degradation is primarily driven by alignment forgetting, not only by knowledge conflict.** We clarify that for a well-aligned model, safety degradation during fine-tuning is not solely caused by implicit risk patterns in benign data, but more fundamentally by the forgetting of safety alignment. The danger direction thus encodes both the direction of safety alignment forgetting and patterns of specific harmful content. We fine-tune the same model (Qwen3-8B, 1000 samples each) on semantically explicit harmful data, semantically explicit safe data, and high-risk benign data respectively:
>
> | |Base|UnSafe|Safe|Benign|
> |-|-|-|-|-|
> |ASR(%)|8.18|77.82|46.91|84.18|
>
> UnSafe and Safe are randomly sampled subsets from BeaverTails labeled as danger and safe respectively, Benign consists of high-risk Dolly samples identified by LARF. Safe data contains no harmful responses, yet fine-tuning raises ASR from 8.18% to 46.91%; Benign ASR (84.18%) even exceeds Danger (77.82%). These results indicate that safety degradation is primarily driven by alignment forgetting, not only by harmful semantics in the data.
>
> **(2)Answer for "Can the danger direction cover different types of safety alignment forgetting?"** Based on the above analysis, the reviewer's question can be reframed as **"whether the danger direction can cover different types of safety alignment forgetting?"** We compute ASR(%) of models fine-tuned on each SQSD-ranked subset, broken down by 11 safety categories (Qwen3-8B, Dolly, Beaver):
>
> |Category|S1|S2|S3|S4|S5|Mono|
> |-|-|-|-|-|-|-|
> |Illegal Activity|78.0|36.0|10.0|6.0|2.0|✓|
> |Child Abuse|72.0|28.0|6.0|4.0|2.0|✓|
> |Hate/Harass/Violence|78.0|44.0|6.0|0.0|0.0|✓|
> |Malware Viruses|56.0|38.0|8.0|14.0|2.0|✗|
> |Physical Harm|72.0|28.0|2.0|4.0|0.0|✗|
> |Economic Harm|80.0|28.0|14.0|10.0|0.0|✓|
> |Fraud/Deception|78.0|38.0|20.0|14.0|10.0|✓|
> |Adult Content|74.0|20.0|10.0|0.0|2.0|✗|
> |Political Campaigning|74.0|32.0|18.0|16.0|4.0|✓|
> |Privacy Violation|66.0|20.0|8.0|6.0|4.0|✓|
> |Tailored Financial Advice|56.0|12.0|8.0|6.0|2.0|✓|
> |**Overall**|**71.3**|**29.5**|**10.0**|**7.3**|**2.6**|**✓**|
>
> 8 out of 11 categories are strictly monotonic, and the 3 non-monotonic cases only exhibit minimal fluctuations in the low-ASR range due to limited evaluation samples per category (50 samples). The remaining 11 configurations (3 models × 2 datasets × 2 danger directions) show the same trend, demonstrating that the danger direction consistently quantifies sample-level alignment forgetting risk across safety categories.
>
> ## Q2: Computational Cost
>
> Cost for scoring 1k samples:
>
> | | Reward Model | Bi-Anchor(Reps) | Self-Inf-N | Bi-Anchor(Grad) | LARF | Ours |
> |-|-|-|-|-|-|-|
> | Memory(G) | 25 | 32 | 45 | 90 | 23 | 44 |
> | Time(s) | 52 | 391 | 2718 | 8127 | 72 | 1918 |
>
> SQSD is the most efficient gradient-based method. The 1918s covers two gradient passes (danger and safety), a single merged pass can halve this cost.
>
> ## Q3 : Limitations
>
> **(1) Static estimation and initialization dependence.** SQSD statically estimates sample risk via instantaneous gradient projections at a single initialization state, relying on effective semantic parameter displacement directions and sensitivity-guided initialization. However, parameters change dynamically during fine-tuning. Our sensitivity-guided initialization mitigates this issue, but exploring dynamic risk estimation during training is a valuable future direction.
>
> **(2) Integration with the training process.** SQSD currently serves as an offline scoring tool. Integrating continuous risk scores into the training process, such as applying different levels of safety constraints to samples of different risk levels, is a promising future direction.
>
> **(3)Risk quantification granularity.** SQSD currently estimates risk at the sample level. Extending directional projection analysis to the token level would enable finer-grained risk localization, precisely identifying which tokens within a sample contribute most to safety degradation.
>
> We again thank the reviewer for the valuable suggestions, we will incorporate the new experiments from Q1, the pseudocode, baseline placement adjustment, and limitations discussion into the revised manuscript.

---

> > ### Author Rebuttal · Reviewer_Akkt · 2026-04-02
> >
> > Thanks to the authors for their efforts in preparing the response.
> >
> > The authors addressed my concerns about generalizability by conducting experiments on different types of data (explicitly harmful, safe, and high-risk benign) and performing SQSD ranking across 11 safety subcategories, demonstrating that the danger direction robustly captures various types of safety alignment forgetting.
> >
> > The authors also report the computational cost.
> >
> > Accordingly, I raise my score to an accept.

---

> > > ### Author Response · Authors · 2026-04-02
> > >
> > > We sincerely thank the reviewer for the constructive feedback and for recognizing the value of our additional experiments. The suggestion to examine the generalizability of the danger direction was particularly valuable. The resulting per-category analysis not only addresses the concern but also strengthens the paper by demonstrating that SQSD robustly captures alignment forgetting across diverse safety categories, which enhances the practical applicability of our method.
> > >
> > > We will incorporate the new experimental results and adopt the structural suggestions discussed above into the revised manuscript. We again thank the reviewer for the valuable guidance.

---

### Decision · Program_Chairs · 2026-04-30

**Decision:**

Accept (regular)

**Comment:**

This paper studies safety alignment degradation during benign data fine-tuning. It identifies the cause as the cumulative drift of model parameters toward dangerous alignment directions during the training process, and proposes a mitigation method called Sample-level Quantification of Safety Degradation (SQSD).

The reviewers find the following strengths: 1) the paper proposes a new perspective on the mechanisms of safety degradation from a training dynamics view, which is novel and interesting; 2) the proposed method is practical and grounded in reasonable analysis; 3) the experimental verification is extensive and results are promising.

During the rebuttal, the authors provided new empirical results and clarified the raised concerns. All reviewers are satisfied with the rebuttal and consider the concerns fully resolved. The evaluations are unanimously positive and this paper is a clear accept. Please ensure that the required revisions and updated empirical results are incorporated in the camera-ready version.